# All-polymer piezo-ionic-electric electronics

Tianpei Xu[1], Long Jin [1] ✉, Yong Ao[1], Jieling Zhang[1], Yue Sun[1], Shenglong Wang[1], Yuanxiao Qu[1], Longchao Huang[1], Tao Yang[1], Weili Deng [1] & Weiqing Yang [1,2] ✉

Piezoelectric electronics possess great potential in flexible sensing and energy harvesting applications. However, they suffer from low electromechanical performance in all-organic piezoelectric systems due to the disordered and weakly-polarized interfaces. Here, we demonstrated an all-polymer piezo-ionic-electric electronics with PVDF/Nafion/PVDF (polyvinylidene difluoride) sandwich structure and regularized ion-electron interfaces. The piezoelectric effect and piezoionic effect mutually couple based on such ion-electron interfaces, endowing this electronics with the unique piezo-ionic-electric working mechanism. Further, owing to the massive interfacial accumulation of ion and electron charges, the electronics obtains a remarkable force-electric coupling enhancement. Experiments show that the electronics presents a high $d_{33}$ of ~80.70 pC $N^{-1}$, a pressure sensitivity of 51.50 mV $kPa^{-1}$ and a maximum peak power of 34.66 mW $m^{-2}$. It is applicable to be a transducer to light LEDs, and a sensor to detect weak physiological signals or mechanical vibration. This work shows the piezo-ionic-electric electronics as a paradigm of highly-optimized all-polymer piezo-generators.

Prompted by the continuous pursuit of new renewable energy and the new generation of wearable devices, the flexible piezoelectric electronics have gained significant momentum in recent years[1–3]. Among them, polyvinylidene difluoride (PVDF)-based piezoelectric electronics, together with flexibility and electromechanical conversion characteristics, caters to the pursuit of smart self-powered sensors in Artificial intelligence Internet of Things (AIoT)[4–7]. However, PVDF is limited to its ceiling piezoelectric coefficient of -33 pC $N^{-1}$, which is still far behind piezoelectric ceramics such as lead zirconium titanium (PZT)[8–11].

To break through the above bottleneck, PVDF is classically added with polar inorganic fillers like ceramic nanoparticles[12,13], carbon nanotubes[14] and MXene nanosheets[15,16]. These nanofillers induce -CH₂-/-CF₂- dipoles orientation for long-range and stable β-phase of PVDF, resulting in speedy upgrading of electronics output[13,16–20]. More importantly, it is revealed that the interfacial polarization effect in PVDF co-blending systems also has a salient positive impact on the electrical output[21–24]. However, the filler-formed second phase often doesn't possess intrinsic force-electric coupling properties. Thus, it is difficult for such piezoelectric electronics to achieve adequate

electromechanical optimization exclusively through such weakly interfacial polarization effect. Additionally, due to the unavoidable trade-off between optimal electrical performance and mechanical properties in above organic-inorganic hybrid systems[25,26], it is necessary to further develop all-polymer piezoelectric electronics.

As a well-known perfluorosulfonate ionomer, Nafion consists of a polytetrafluoroethylene backbone and side chains terminated by sulfonic acids. It relies on group dissociation to achieve proton conductivity with the model of a cluster network structure[27,28]. Stimulated by non-uniform stress, ionomer Nafion undergoes non-uniform internal ion migration, which in turn generates ionic potential differences, referred to the piezoionic effect[29–32]. And clearly, both as piezo-generators, piezoionics and piezoelectrics have similar macroscopic effects on the force-electric response[29,33,34]. Combining the two effects enables the development of self-powered hybrid systems with simple structure and unique mechanisms, will be greatly improved compared to the existing piezoelectric-triboelectric hybrid generators[35–37]. On this basis, the similarity in chain configuration certainly fueled the creation of an all-polymer piezo-generator of PVDF/Nafion with strongly interfacial polarization, as a novel strategy for optimizing

[1]Key Laboratory of Advanced Technologies of Materials (Ministry of Education), School of Materials Science and Engineering, Southwest Jiaotong University, Chengdu 610031, China. [2]Research Institute of Frontier Science, Southwest Jiaotong University, Chengdu 610031, China. ✉e-mail: longjin@swjtu.edu.cn; wqyang@swjtu.edu.cn

electromechanical performance. In fact, the piezoelectric optimization via particle-doped Nafion was previously revealed by PVDF composite poling-free electret[38]. However, the existing disordered ion-electron interface leads to internal charge neutralization losses, preventing the piezoelectric/piezoionic systems from desirable performance. Therefore, the regularized and wide-area ion-electron interface is crucial for unlocking its performance potential, as well as excavating the unusual working mechanism.

Herein, we present an all-polymer piezo-ionic-electric electronics with PVDF/Nafion/PVDF (PNP), featuring ordered and strongly-polarized ion-electron interfaces for remarkable electromechanical performance boost. The sandwich structure was constructed by confining the effect of the directional pressure field and precise thermal field. Benefiting from the massive accumulation of interfacial polarized charges at such ion-electron interfaces, this PNP all-polymer has a high $d_{33}$ of -80.70 pC N$^{-1}$, giving the electronics with a pressure sensitivity of up to 51.50 mV kPa$^{-1}$ and an instantaneous peak output power of up to 34.66 mW m$^{-2}$ (under 177 kPa). What is remarkable is that the PNP electronics achieve piezoelectric self-polarization by bending behaviors, which confers a unique piezo-ionic-electric working mechanism. It is also facilitated by the piezoelectric and piezoionic effects coupling at the ion-electron polymer interface. We further demonstrate the potential applications of PNP electronics for harvesting mechanical energy and detecting vibration frequencies, among others. Leveraging the concept of piezo-ionic-electric electronics, this work opens up a brand-new idea for highly optimizing the electromechanical properties of flexible electronics, expected to promote the valuable utilization of all-polymer piezo-electronics.

## Results

### Concept, design and properties

The concept of piezo-ionic-electric electronics is described in Fig. 1a. As mentioned above, both the piezoelectric effect and the piezoionic effect describe the physical transition from deformation to polarization. The difference is that the former relies mainly on changes in the crystal dipole to produce electron-based polarization[1,39,40], whereas the

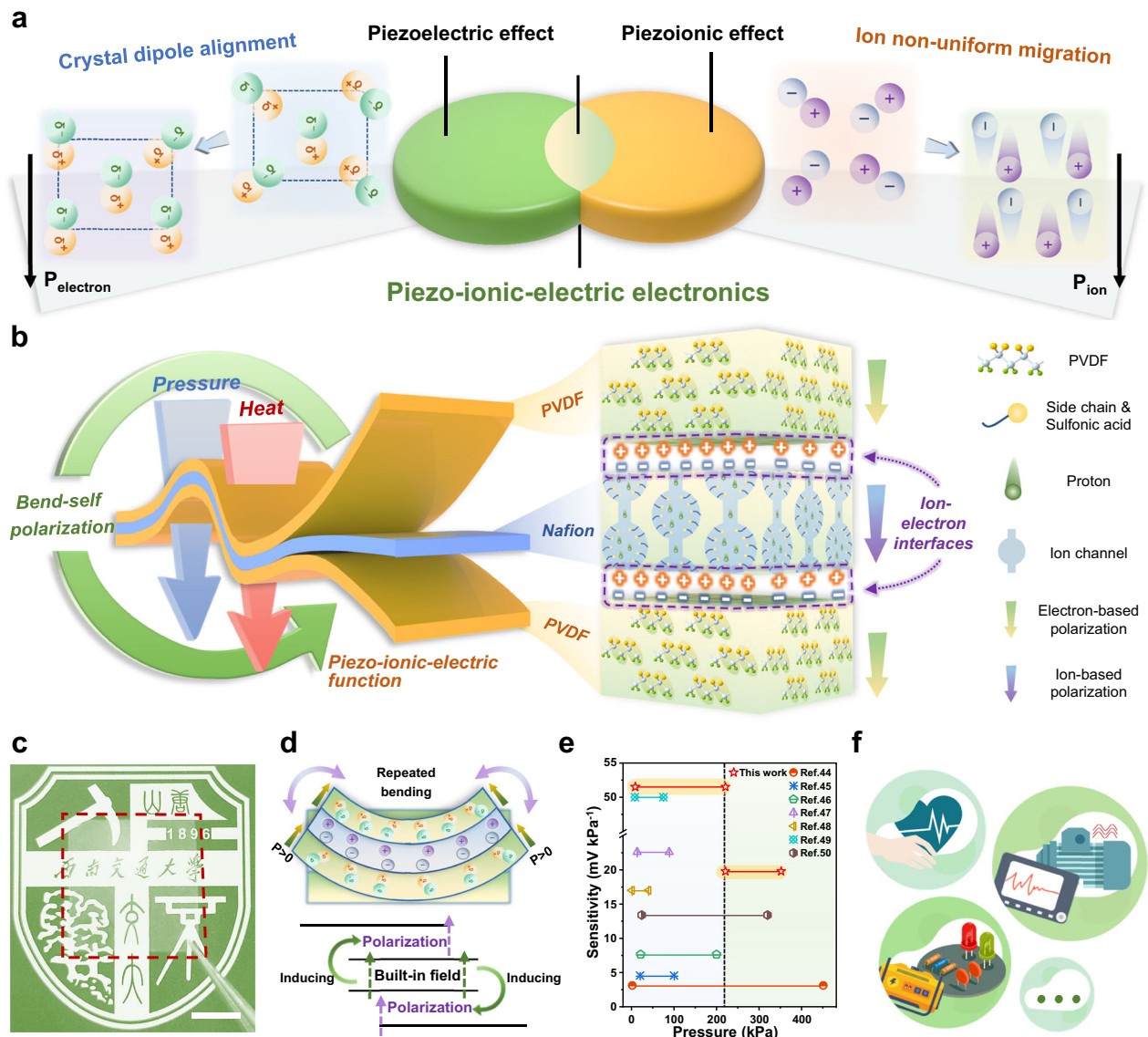

**Fig. 1 | Concept, design and properties of the piezo-ionic-electric electronics.** **a** Concept of piezo-ionic-electric electronics derived from the intersection of the piezoelectric and piezoionic effects. **b** Schematic for the hot-pressing process and the structural basis of the piezo-ionic-electric function. **c** Photograph of the all-polymer transparent PNP film. The scale bar is 10 mm. **d** After repeated bending stimulation, the PNP electronics achieves piezoelectric activation, a particular process called bend-self-polarization. **e** Comparison about pressure sensitivity of our PNP thin-film electronics with existing PVDF-based film-type piezoelectric sensors. **f** Application areas of the piezo-ionic-electric sensors and transducers.

latter produces ion-based polarization originating from the non-uniform migration of ions[29,34,41]. The piezo-ionic-electric electronics stems from the intersection, which highly couples two force-electric conversion principles. In other words, it is defined as a self-powered piezo generator that relies on both the piezoelectric effect and the piezoionic effect to realize the force-electric conversion function. Based on the concept of the novel piezo-system, the material designs are shown in Fig. 1b. For heterogeneous layered hybrid piezo-films, the construction of stable ordered interfaces and the orientation in the out-of-plane direction are crucial for the overall polarization of the electronics[21,42,43]. Therefore, in this study we successfully and efficiently construct PVDF heterogeneous multilayers containing Nafion sandwiched layer using a simple molding process of laminated hot-pressing (Supplementary Fig. 2), preceded by pre-preparation of PVDF (Supplementary Fig. 1) and Nafion films (Supplementary Fig. 3). The whole process is easy to operate and suitable for scaling up to industrialization. Flat and wide-domain ion-electron interfaces of the PNP all-polymer film can be constructed by modulating the confining level of oriented pressure field and the melt-quench thermal field. At the same time, the critical structures on both sides of the interface, including the β-phase effective for piezoelectric effect and the intrinsic protons nano-channels for piezoionic effect, are bound into a functional unity. Fig. 1c shows that it has a homogeneous texture and good transparency, with a thickness of ~50 μm. To further test its electrical properties, dense silver electrodes are sputtered on both sides of the PNP film by physical vapor deposition and mask template methods, and encapsulate with polyurethane (PU) tape to shield the electronics from environmental and friction electrical signal disturbances. Supplementary Fig. 4 and Supplementary Table 1 present the final structure of the prepared flexible PNP piezo-ionic-electric electronics as well as the testing results on the thickness of each layer. Then, by simple repeated bending stimulation, the PNP electronics can achieve self-polarized activation, which in turn enables the piezo-ionic-electric response under vertical stress stimulation (Fig. 1d and Supplementary Fig. 5). This is a unique piezoelectric activation mechanism different from high-voltage polarization, that thanks to the dependence of the piezoionic effect on non-uniform deformation, while bending behavior maximizes the strain non-uniformity of film-materials across the whole thickness range. Thus, the bending behavior can generate brief but frequent built-in ion-based electric fields for piezoelectric activation, which is named as bend-self-polarization. Ultimately, due to the coupling of piezoelectric and piezoionic effect, the accumulation of activated polarized charges at the ion-electron interfaces endow the PNP electronics with highly sensitive electrical response over a broad pressure range, with a comparison[44–50] in Fig. 1e and Supplementary Table 2. Other excellent electromechanical properties will be shown in detail below. This grants the electronics sufficient potential for applications in physiological signal sensing, mechanical vibration monitoring and flexible energy harvesting (Fig. 1f).

## Working mechanism

As mentioned previously, we are delighted to find that the PNP electronics does not require additional corona-polarization after preparation to activate piezoelectric response as before. What replaces such high-voltage process is the simpler stimulation through repeated bending in the same direction (bending moment direction). Moreover, a large number of samples have shown statistically that the number of bending required to achieve polarization exhibits a negative correlation with the bending curvature (Supplementary Fig. 6, details in Supplementary Note 1). At approximate limiting curvature (0.332 mm$^{-1}$), the average number of bending is about 32 times. In addition, the X-ray diffraction (XRD) spectra show (Supplementary Fig. 7a) that the PNP film has a sharp diffraction peak at 20.1°, mainly corresponding to the (200) crystal plane of the β phase[11], which is clearly different from the blunt peak of the pre-prepared coated film (CP). It proves that the PVDF

layer undergoes crystalline reorganization, from predominantly amorphous phases to a more complete crystalline structure, induced by the directional pressure with the thermal process. Further, As given in Supplementary Fig. 7b and c, Fourier transform infrared (FTIR) spectra shows that the PNP film contains a certain amount of β-phase whose characteristic peak appears at 840 cm$^{-1}$ wavenumber[42], and the relative content is higher at a lower pressure of 100 MPa, with 41% (see Supplementary Note 2 for calculation method). This data is similar to that of the PVDF four-layer film (36%) obtained by "pressing-and-folding", thus supporting the rationality[42].

Based on such phenomenon and structure, we propose the piezo-ionic-electric working mechanism for PNP electronics, illustrated in Fig. 2a. Stage I is the piezoelectric activation by bend-self-polarization. Element a(i) shows the initial state of the PNP electronics obtained by laminated hot-pressing. The β-phase within the upper and lower PVDF layers is abstracted as dipoles disorderly arranged in the out-of-plane direction (polarization P ≈ 0), even though most of them tend to be oriented under the constraint of the pressure and thermal fields. Also, the sulfonate groups and free protons of Nafion interlayer are in charge equilibrium, abstracting as negative ions (main chain and side chains with sulfonates) and positive ions (protons). Subsequently, the bending stimulation of the film electronics induces drives the dissociated protons to move directionally in the nanochannels toward the in-plane expansion side, which stems from the piezoionic effect of Nafion. Repeated bending stimulations with the same bending moment direction induce an ion-induced electric field in the intercalation. In turn, the built-in electric field leads to intense ion-electron interactions between the two ion-electron interfaces. It amplifies the polarization induction, promoting the internal alignment and overall orientation (spontaneous polarization P$_s$ > 0) between the dipoles of the two PVDF layers, as depicted in Element a(ii). Finally, Element a(iii) shows the completion of the self-polarization process: the ions at the intercalation are uniformly distributed but more easily migrated, while the upper and lower dipoles achieve consistent polarization alignment after molecular chains relaxation (residual polarization P$_r$ > 0). This is followed by Stage II, where the PNP produces dynamic piezoionic-coupled piezoelectric response. When the activated Element a(iv) is strained, the dipole polarization within the PVDF layer increases, generating a piezoelectric electric field. And Nafion undergoes ionic polarization based on the piezoionic effect under the constraint of this interfacial electric field, ultimately achieving an overall ion-electron dual-polarization. Meanwhile, the top and bottom electrodes generate opposite charges in the presence of this overall electric field, and charge transfer occurs via an external circuit, which in turn generates an electrical signal. Element a(v) describes the above process. When the strain reaches a maximum, the overall polarization ceases to change and there is no more charge transfer in the external circuit, as described by Element a(vi). Element a(vii) shows that after the stress is released, the strain begins to recover and the degree of polarization begins to change in the reverse trend (decrease), allowing an equal amount of charge to move in the reverse direction via the external circuit. Eventually, the strain is fully recovered and the PNP returns to its initial state, completing the cycle of dynamic piezoionic-coupled piezoelectric response.

In order to justify the above piezo-ionic-electric working mechanism, we designed two sets of validation experiments. At first, the rationality of bend-self-polarization is explored by decoupling the relationship between the direction of the bending moment and the phase of the short-circuit current. As in Fig. 2b, we first prepared two identical PNP electronics named PNP-A and PNP-B respectively. To distinguish the bending direction, a Z-axis uniaxial coordinate is established in the direction perpendicular to the plane of the film. Bending stimuli were applied to PNP-A and PNP-B in the +Z and −Z directions, respectively, to achieve piezoelectric activation. Thereafter, current profiles were acquired using a piezoelectric test system

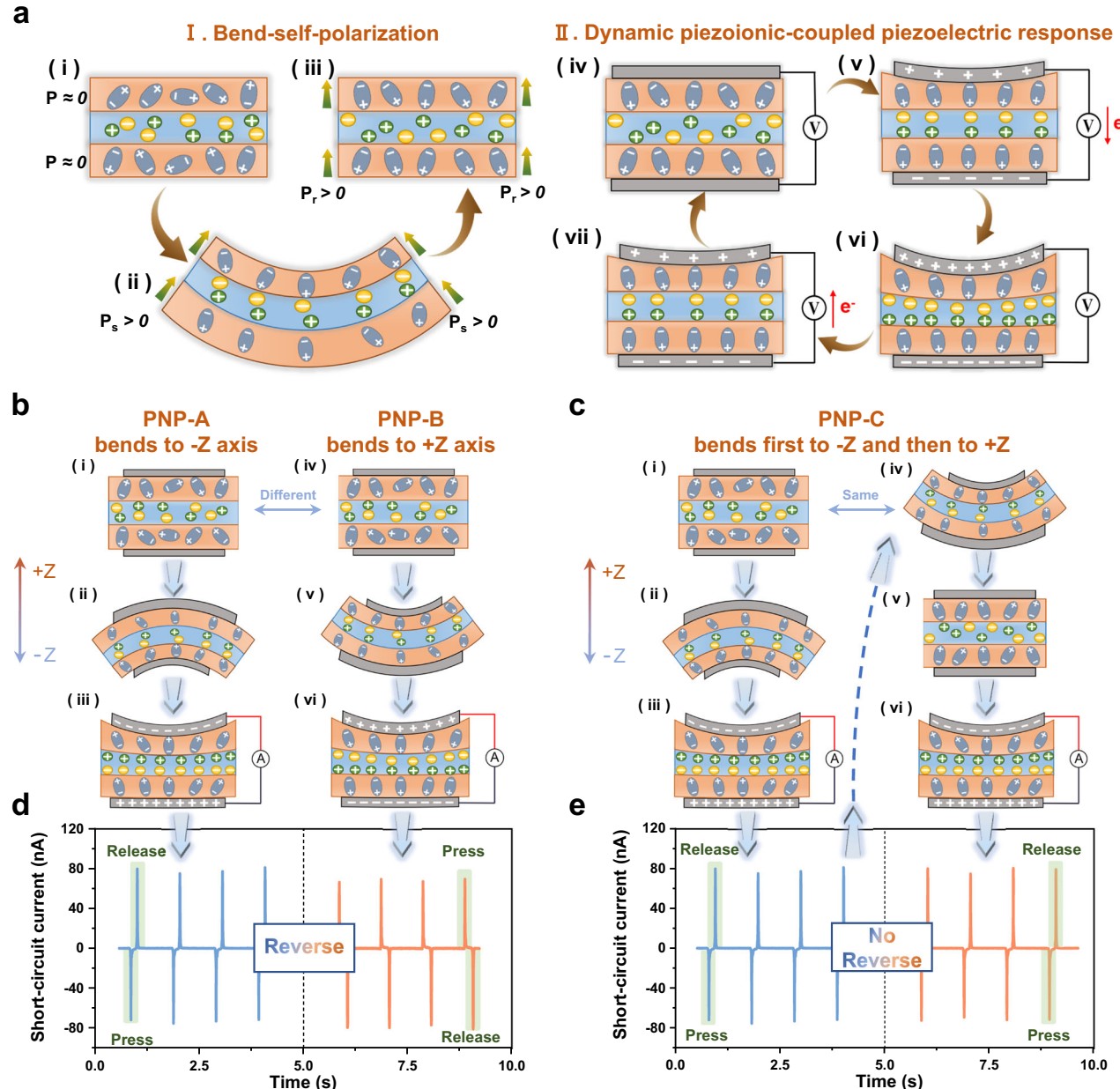

**Fig. 2 | The unique piezo-ionic-electric working mechanism of the PNP electronics and its verification. a** Schematic diagram of the two stages (seven elements in total) in the working process of the PNP electronics, abstractly depicting the microstructural evolution inside the PNP. **b** The process and principles of the validation experiments in Stage I. When the two PNPs were bent in opposite directions (−Z and +Z), the difference of internal microscopic evolution are compared schematically. **c** The process and principles of the validation

experiments in Stage II. For the same PNP, bending in one direction (−Z) was followed by bending in the other direction (+Z), whose micro-evolution is also shown. **d** Plot of opposite short-circuit current phases, supporting the decisive role of the piezoionic effect in the bend-self-polarization. **e** Plot of unchanged short-circuit current phase, reflecting the leading role of piezoelectric effect in the overall response process.

as in Supplementary Fig. 8. The test results are shown in Fig. 2d. It is clear that the two devices output opposite piezoelectric phases during each "press-release" stress cycle. For PNP-A, the current first goes down when pressed and then goes up when released. In contrast, for PNP-B, the current goes up before going down. Considering that the consistency of the test port connections was tightly controlled during testing, this difference in phase can only be attributed to differences in activation conditions (bending direction). In detail, under different bending directions, the piezoionic effect of Nafion induces two ion-based electric field of opposite polarity, which in turn induces the dipoles in the PVDF to realize completely opposite overall orientations, and hence the subsequent generation of opposite piezoelectric

phases. Element b(i) ~ b(vi) microscopically depict the differential processes described above. Thus, it is the piezoionic effect that plays a decisive role in Stage I. Further, we design a similar validation experiment as in Fig. 2c. Only one PNP electronics, named PNP-C, is bent in the −Z direction to activate piezoelectricity, and the phase of the current is tested. Subsequently, bending this film in the opposite direction (+Z direction) repeatedly and testing whether the phase changed (the interface connections remained consistent). The results (Fig. 2e) show that there is no change in the phase before and after the reverse bending, as well as no difference in the magnitude of the short-circuit current. This demonstrates that for the same PNP electronics, the PVDF polarization direction determined in Stage I is stable or does

not change readily with the ion-induced electric field. It is worth mentioning that if PVDF is replaced with polytetrafluoroethylene (PTFE), a non-piezoelectric polymer, the piezoelectric response after bending self-polarisation cannot be achieved (Supplementary Fig. 9). Thus, they further justify that the piezoelectric effect controls the piezoionic effect in Stage II. In addition, we supplemented the short-circuit phase calibration experiments (Supplementary Fig. 10 and Supplementary Note 3 for details). According to the experimental results, for the PNP device, the tensile side during bend activation can be equated to the positive pole of the power (when pressurized). This is consistent with Nation's theory of piezoionic effect[29]. It further validates the piezo·ionic·electric mechanism and the respective roles of these two effects in it.

Here, this unique piezo·ionic·electric working mechanism is discussed and preliminarily demonstrated. And it essentially originates from the stable and strong interaction between electrons and ions at the heterogeneous interfaces, which is also the core of this interfacial ionic-electronic effect.

## Electromechanical enhancement

Based on the above working mechanism, this piezo·ionic·electric electronics exhibits enhanced electromechanical properties. The successful construction of stable ion-electron polymer-interfaces in PVDF/Nafion/ PVDF multilayer heterostructure is the basis for obtaining excellent force-electric coupling properties. Molecular dynamics (MD) simulation can reveal the binding and induction of PVDF/Nafion based the interfacial interaction (see Supplementary Note 4 for details). The bilayer-polymer interface model of 30 PVDF molecular chains and 20 Nafion molecular chains was constructed and MD calculations were implemented. After a 1000 ps interaction, the two are closer together, while the out-of-plane polarization of the PVDF chains undergoes a reversal and increase, with 14.80 D in -Z direction (Fig. 3a). In fact, this molecular dynamic process shows a reversal trend in the change of the out-of-plane polarization of PVDF chains (Supplementary Fig. 11), which is consistent with the trend of interaction between the $-CF_2$ moieties of Nafion and the $-CH_2$ moieties of PVDF. Furthermore, by analyzing the dihedral angle changes in the conformation of PVDF chains during different dynamic time periods, it is clear that the dihedral angle fraction near 180° (*trans* or T conformation)[51] gradually increases (Fig. 3b). This indicates the interfacial interaction induces a tendency of all-*trans* polar conformation (β phase) transform, which forms the conformational basis for bend-self-polarization and electromechanical enhancement. Also, Fig. 3c shows the radial distribution functions (RDFs) of a large number of F atoms of Nafion close to the interface and all H atoms of PVDF backbones. Apparently, the two types of atoms are closer to each other after 1000 ps compared to the beginning of the dynamics (1 ps) because of interaction, which facilitates the stable formation of the polymer interface and provides a suitable distance for hydrogen bonds. Calculated by MD simulation, the energy of such interfacial interaction first increases and then equilibrates after about 1000 ps, exhibiting a large interaction energy (-950 kcal mol$^{-1}$) (Fig. 3d). Meanwhile, it is worth noting that this energy originates from hydrogen bonding interactions as well as dipole interactions occurring at the interface between Nafion and PVDF.

Likewise, the cross-sectional SEM image in the Fig. 3e confirms that tight binding of the PVDF upper and lower layers (orange area) and the Nafion interlayer (blue area). Moreover, FTIR in transmission mode of the three films, PNP, PPP (hot-pressed PVDF three-layer film, see Experimental section for details), and Nafion, also confirm the build-up of heterogeneous interfaces (Fig. 3f). The spectra of the fingerprint region show that in the 475-650 cm$^{-1}$ wavenumber range (especially in the circular dashed box), the PNP peaks contain both PPP and Nafion peaks, with a stacking effect.

In order to visualize the enhancement effect of the introduction of the Nafion interlayer on the overall electromechanical performance,

PPP and Nafion are selected as the control group to compare the electrical output with PNP electronics. Note that PPP experienced an electrical-poling process beforehand, as detailed in the experimental section and in the Supplementary Fig. 12. Under the same test conditions of pressure excitation (88 kPa) as well as electrode area (1.13 cm$^2$), the comparative plots of open-circuit voltage ($V_{OC}$) and short-circuit current ($I_{SC}$) of the three are shown in Supplementary Fig. 13a and b, respectively. It's clear that the $V_{OC}$ and $I_{SC}$ of PNP are as high as 5.45 V and 370 nA, both of which are substantially elevated compared to the control groups. Additionally, we have compared the output of the PNP roughly with a polarization-prepared commercial PVDF piezoelectric film electronics (from PolyK company). The result in Supplementary Fig. 14 also shows that our PNP electronics has a superior output obviously. Further, the $d_{33}$ of PNP was measured by the direct piezoelectric charge method (in Supplementary Note 5) and the results are shown in Fig. 3g. The slope of the transfer charge-force fitted function is calculated as $d_{33}$[9,12]. The $d_{33}$ of up to 80.70 pC N$^{-1}$ is much higher than that of PPP, and the outstanding value is 2.6 times higher than that of polarization-prepared commercial PVDF, reflecting its highly-optimized force-electric coupling performance. All samples were repeated three times (Supplementary Fig. 15), and then averaged to avoid chance. In addition, it is necessary to exclude the interference of triboelectric charges during the testing process[52]. We used the recently proposed compressed balance analysis method[53] as a correction for piezoelectric charge testing. The results show that the $d_{33}$ value of the two methods are quite similar (For details, see Supplementary Note 6 and Supplementary Fig. 16). It is worth noting that the measured $d_{33}$ of commercial PVDF is 31.10 pC N$^{-1}$, which is within the range of values advertised by the product, laterally corroborating the reasonableness of this $d_{33}$ measurement method. In fact, as shown in Supplementary Fig. 17 and Supplementary Table 3, the $d_{33}$ of PNP is at the leading edge of PVDF-based piezoelectric systems, with or without additional electrical polarization.

Subsequently, for exploring the intrinsic reasons for this performance enhancement, the β-phase content of the above PVDF-based films is comparatively analyzed first. The preceding XRD spectra (Supplementary Fig. 7a) shows the crystalline transformation resulting from the hot-processing. For further quantitative analysis, the crystallinity ($X_C$) of the three films is tested with differential scanning calorimetry (DSC) (Supplementary Fig. 18a). The β-phase ratio within the crystal region (F(β)) is analyzed using FTIR in attenuated total reflectance (ATR) mode (Supplementary Fig. 18b). The final calculated net β-phase crystal contents of PNP and PPP are approximately the same (12.57% and 12.54%), while CP contains almost no β-phase crystals (detailed calculations are shown in the Supplementary Note 2). So, the introduction of Nafion interlayer will not affect the positive effect of hot-pressing method on the β-phase transformation of PVDF. More importantly, an important conclusion can be drawn that the electromechanical boost of the PNP is not a result of the change in β-phase content. At the same time, we speculate that such boost is related to the clustering of ions and electrons in the interface region, again as the positive influence of the interfacial ionic-electronic effect. Fig. 3h–j show the macroscopic composition of the PPP, Nafion, and PNP devices, respectively, as well as the internal piezoelectric/piezoionic sensing structural units. For the PPP, the interior is nearly homogeneous without interfaces, although it has undergone laminated hot-pressing. Therefore, like pure PVDF films, it relies only on the dipoles transition of the PVDF polar phase (not much of β-phase for PPP) to realize the piezoelectric effect, and thus has a smaller polarized electric field and less output (Fig. 3h). Similarly, pure Nafion ion-device (after repeated bending stimulation) is able to rely on the migration of protons dissociated from sulfonate groups in the nanoscale ion channels to realize piezoionic effects under stress, generating an ionic-induced electric field, and thus also outputting electrical signals (Fig. 3i). Unlike them, PNP has wide-domain interfaces in sandwiched

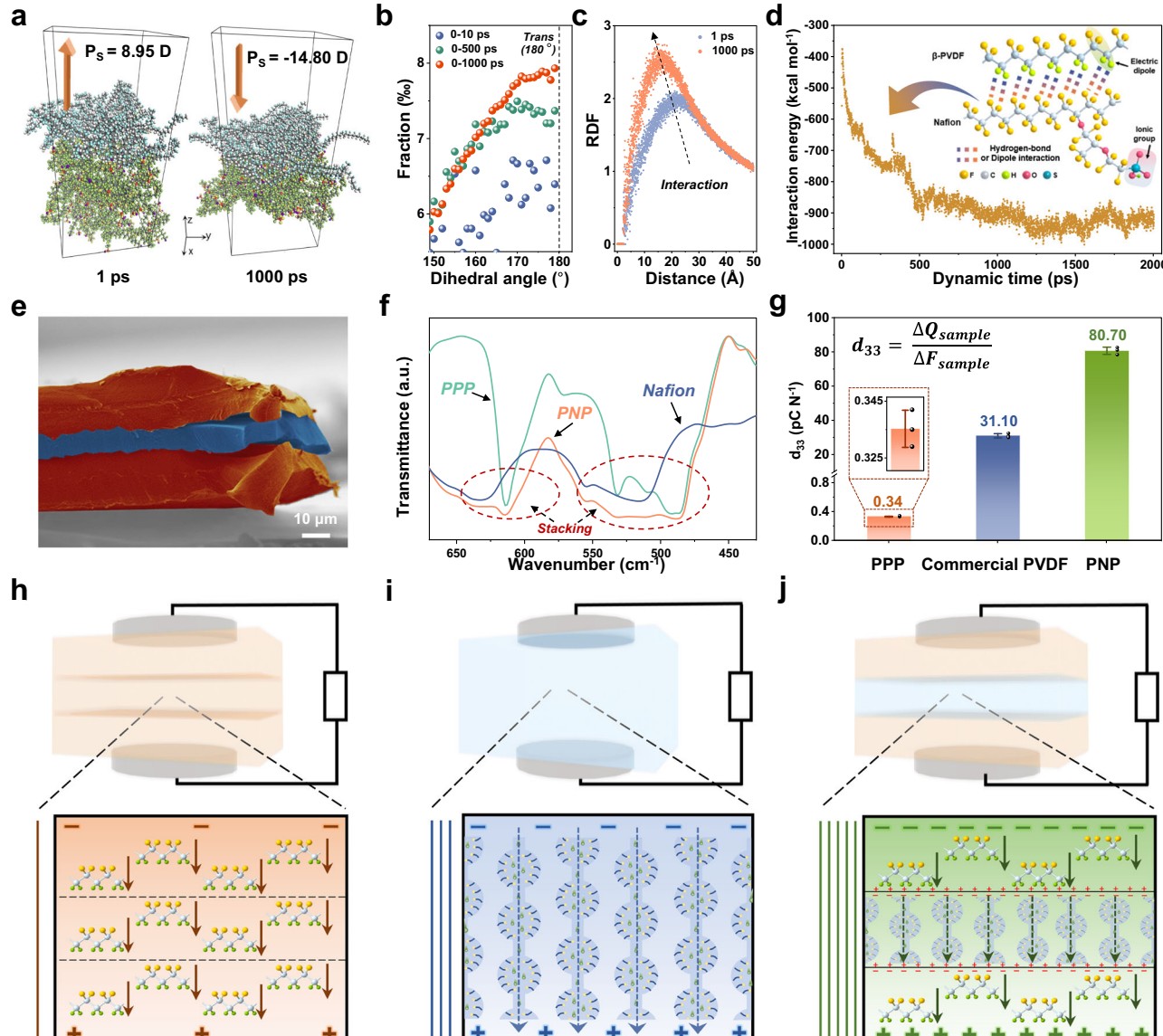

**Fig. 3 | Structural basis and performance comparison of PNP for achieving electromechanical enhancement. a** The initial (1 ps) and final (1000 ps) snapshots for molecular dynamics (MD) simulations of the bilayer-polymer interface model of PVDF and Nafion chains and the corresponding polarization states. **b** Distributions of PVDF chains dihedral angles near 180° (*trans* or T) in three increasing time periods. **c** Radial distribution functions (RDFs) between F atoms of Nafion close to the interface and all H atoms of PVDF. **d** Interaction energy evolution for the PVDF layer and Nafion layer over a dynamic time of 2000 ps. The inset shows a simplified schematic of the interaction between PVDF and Nafion. **e** The cross-section SEM image of the PNP (after coloring). **f** Comparison on FTIR spectra of PNP, PPP, and Nafion under transmission mode. **g** Comparison of $d_{33}$ derived from the direct piezoelectric charge method for PPP, commercial PVDF and PNP. Data are presented as mean values ± standard deviations (SDs). Bar heights indicate mean values and error bars indicate SDs, both from the results of three repeats on each sample. Comparison on the macro-structural composition and micro-functional units of **h** PPP, **i** Nafion and **j** PNP devices.

composite structure, which combines the piezoelectric effect of PPP and the piezoionic effect of Nafion, and exhibits a distinct self-powered sensing mode with remarkable electromechanical enhancement effect, that is piezo-ionic-electric sensing described above. The key to such performance improvement lies in this ion-electron polarized interface, where stress-induced polarization is significantly enhanced (Fig. 3j). In detail, under stress, a large accumulation of polarized charges (electron-state) originating from piezoelectric effects occurs at the upper and lower interfaces near the PVDF layers. Correspondingly, an equal amount of polarized charges (ion-state) originating from piezoionic effect accumulates at the two interfaces close to the Nafion interlayer. Unlike the conventional interfacial polarization in dielectric systems, a large number of free-moving ions in the Nafion interlayer participate in that polarization[34], not just the few electron

charges confined within crystals of PVDF. Thus, the difference in dielectric properties of the two components is amplified with the help of the difference in free charge carriers, which in turn induces a higher interfacial charge density and improves the overall charge output. As shown in Supplementary Fig. 19, the charge distribution states of the above three-layered models after potential generating were simulated correspondingly using COMSOL Multiphysics (assuming they are of equal thickness). It can be clearly visualized that PNP has a strongly polarized heterogeneous interface and a greater overall polarization output (Supplementary Fig. 19a).

## Excellent force-electric coupling
As previously discussed, piezo-ionic-electric working mechanism based on interfacial ionic-electronic effect enables the PNP electronics

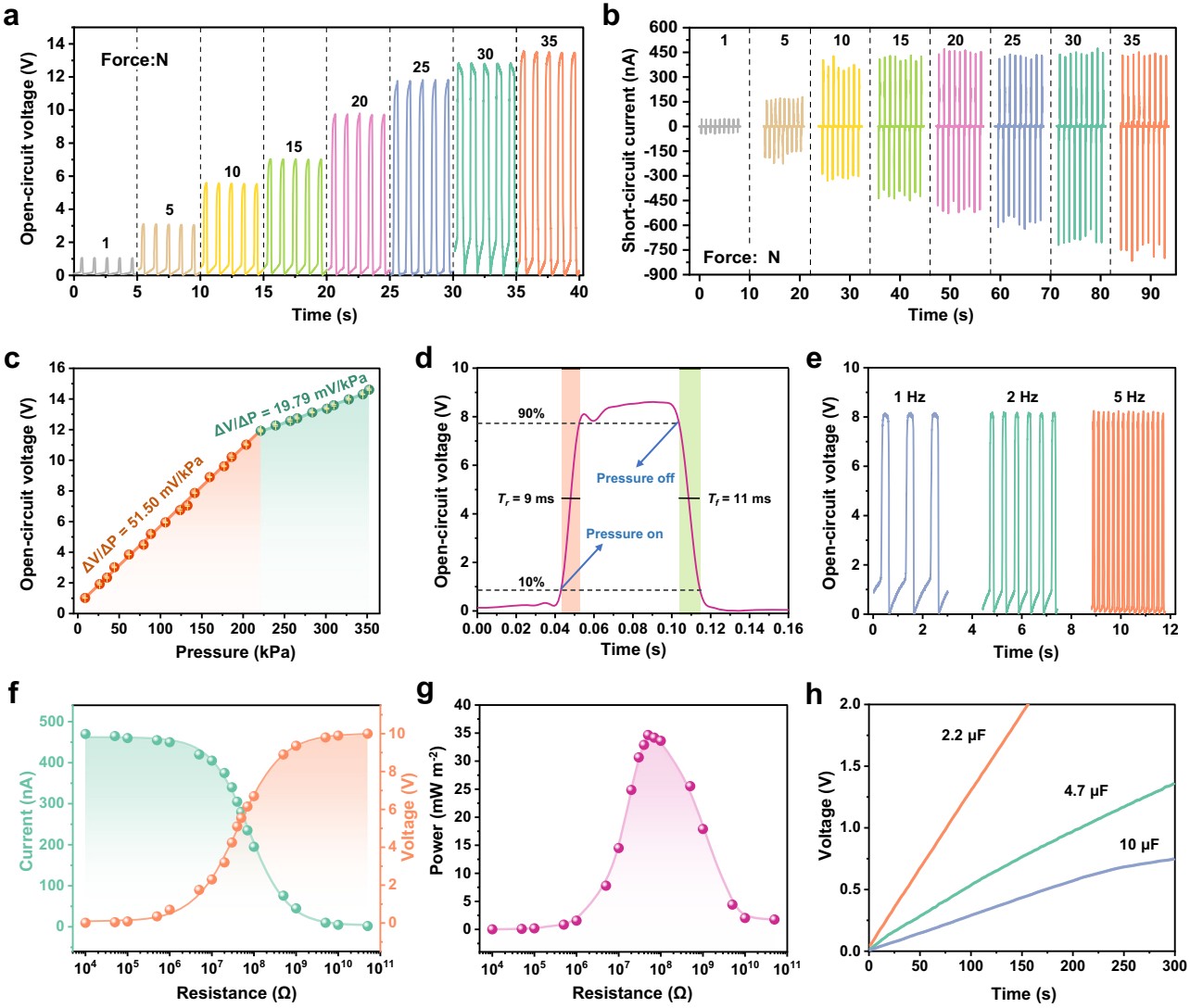

**Fig. 4 | Systematic testing and analysis on PNP piezo-ionic-electric sensing and energy harvesting characteristics. a** Open-circuit voltage at different stresses. **b** Short-circuit current at different stresses. **c** Dependence of open circuit voltage on pressure (from 9 kPa to 354 kPa). Data are presented as mean values ± SDs. Points indicate mean values and error bars indicate SDs, both from the statistic results of three repeats on the same sample. **d** Pressure response time at 5 Hz pressure frequency. **e** Piezo-ionic-electric output at different frequencies (1 Hz, 2 Hz and 5 Hz). **f** Dependence of output current and voltage on external resistance. **g** Instantaneous peak power per unit area with external resistance. **h** Voltage curves when charging different capacitors.

excellent output performance. Here, we tested and analyzed its comprehensive force-electric coupling characteristics. First, electrodes of the PNP electronics were connected to the electrometer first in the forward direction and then in the reverse direction, and it was found that the current also shifted in phase and have almost the same peak current (Supplementary Fig. 20). Further, there is no phase difference between the force loading curve and the electrical signal curve in the same time domain, even in contact-separation testing mode (Supplementary Note 6 and Supplementary Fig. 21). They prove that the generated signal belongs to piezoionic-coupled piezoelectric signal rather than other interfering signals. The voltage and current output of the PNP under gradient pressure (1-35 N, contact-separation mode) were subsequently tested. As seen in Fig. 4a and b, the voltage and current both increase markedly with increasing applied pressure, and the waveforms and peaks are very stable. At a maximum pressure of 35 N (310 kPa), the PNP is capable of outputting 13.5 V and 810 nA, which represents its advanced level. Meanwhile, the testing of the voltage output under several pressure conditions reveals that the PNP exhibits a good linear relationship between pressure and voltage in both the low (9-221 kPa) and high (221-354 kPa) pressure ranges

(Fig. 4c). Defining pressure sensitivity ($S_V$) as the slope of the output voltage-pressure curve ($S_V = \Delta V/\Delta P$, $\Delta P$ is the relative change in the applied pressure and $\Delta V$ is the relative change in the open-circuit voltage). Then the pressure sensitivity of the PNP can be calculated to be 51.50 mV kPa⁻¹ and 19.79 mV kPa⁻¹ respectively, which is also a considerable performance capability. The limited protons content in the Nafion interlayer constrains the saturation tendency of the interfacial polarization, which may be the main reason for the segmented sensitivity of the PNP. Figure 4d illustrates the voltage signal profile in a working cycle (17.5 N, 5 Hz), from which it can be obtained that the PNP has a remarkably fast pressure response time, $T_r/T_f$ of 9/11 ms. Then, the magnitude of voltage output at different frequencies (1 Hz, 2 Hz and 5 Hz) is compared, reflecting its good frequency stability (Fig. 4e). Additionally, the press-release cycle test highlights the excellent long-term signal stability of PNP. During 20,000 cycles of high-frequency external force, the output current fluctuates only slightly and remains stable overall (the output increases slightly after high cycles due to continuous surface charge accumulation) (Supplementary Fig. 22).

Besides the sensing characteristics, we have also briefly investigated the energy harvesting characteristics of the PNP. Fig. 4f

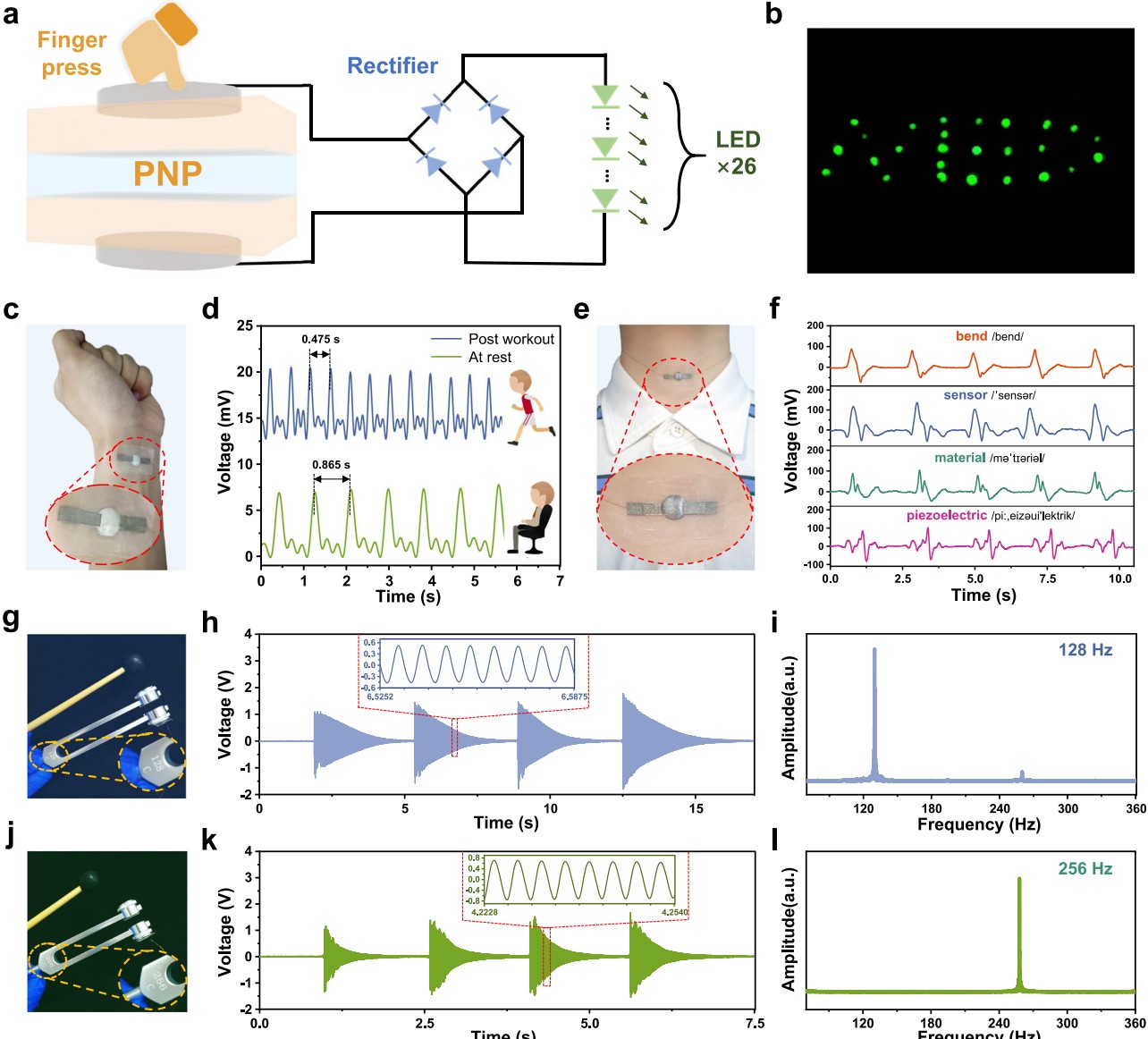

**Fig. 5 | Multifaceted applications of PNP piezo-ionic-electric electronics. a** The PNP device as a transducer for lighting 26 LEDs in series after rectification. **b** Photograph of the illuminated LEDs displaying the word "NED". **c** The PNP electronics attached to the wrist. **d** Pulse wave signals monitored by the PNP at rest and post workout. **e** The PNP electronics attached to the throat. **f** Different voltages signals waveforms generated by the PNP due to differential vibrations of the vocal cords (not on the same timeline). **g** The PNP attached to the tuning fork of 128 Hz, and the detected **h** time-domain plot and **i** frequency-domain plot (after FFT). **j** The PNP attached to the tuning fork of 128 Hz, and the detected **k** time-domain plot and **l** frequency-domain plot (after FFT).

illustrates the output voltage and current of the PNP with different external resistances (20 N, 1 Hz). It can be seen that when the load resistance is increased to 5 MΩ, the current begins to drop sharply, while the voltage begins to rise rapidly. Until after 500 MΩ, the two curves flatten out. The resulting instantaneous peak power curve (see Supplementary Note 7 for the calculation method) is plotted in Fig. 4g. The curve exhibits a typical increasing and then decreasing trend, achieving a maximum peak power of 34.66 mW m⁻² at the resistance of 50 MΩ (50 MΩ can be approximated as the internal resistance of the device). As a comparison, the piezoelectric output under external resistance of the PPP and its instantaneous peak power were also tested, and the results are shown in Supplementary Fig. 23a and b. The results show that the internal resistance and maximum peak power of the PPP device are 100 MΩ and 0.057 mW m⁻², respectively. Obviously, the PNP reduces the internal resistance while increasing the output power by about 600 times, which greatly optimizes the energy harvesting efficiency of thin-film piezoelectric electronics. Therefore, the electrical energy generated by the PNP under the above excitation can charge the capacitor (after rectification), and the voltage curves during charging are shown in Fig. 4h. This demonstrates the potential of PNP to power small electronic devices.

**Piezo-generator applications**

In the light of the many outstanding electromechanical properties of PNP, we have explored their potential for a wide range of applications. Firstly, benefiting from the outstanding power output compared to other normal piezoelectric film electronics, PNP can be used as a transducer to power small LED lights. An experimental circuit diagram was designed as shown in Fig. 5a (Supplementary Fig. 24 shows the object). It has been proved that the PNP electronics was able to light up 26 series-connected LEDs under finger-pressing conditions after connecting a rectifier bridge (Fig. 5b, Supplementary Video 1). In combination with the above performance of PNP for capacitor charging, this

demonstrates the prospective application of PNP, a micron-sized film device, in mechanical energy harvesting.

Secondly, the advantage in sensitivity of PNP endows it with the ability to detect weak mechanical signals. As an example of detecting subtle physiological signals, PNP sensors are used for human pulse wave monitoring, which is significant in preventing cardiovascular diseases such as atherosclerosis. The PNP sensor is mounted on the back of a person's wrist to detect weak pulse waves (Fig. 5c). Whether at rest or post workout, arterial pulses show their waveforms in the form of output voltages (Fig. 5d, Supplementary Video 2). Moreover, the detailed waveform of a single pulse clearly highlights the percussion waves (P-wave), tidal waves (T-wave), and dicrotic wave (D-waves) in the human pulse (Supplementary Fig. 25a and b). The times difference between neighboring peaks (0.865 s and 0.475 s) can be converted to heart rate in the two states, which are 69 and 126 beats per minute, respectively. Such performance exhibits the usability of PNP thin film sensors in real-time physiological monitoring. On the other hand, the PNP is also equipped to monitor the physiological signal of vocal cord vibration. As shown in Fig. 5e, the PNP sensor is attached to the human throat. When the person utters different English words, the PNP detects different vocal cord vibration modes, which in turn produce output voltage signals with different waveforms (Fig. 5f, Supplementary Video 3). It is clear that as the number of syllables in a word increases (from "bend", to "sensor", to "material", and finally to "piezoelectric"), the voltage waveform is complicated and multi-peaked accordingly. Therefore, the potential of PNP in detecting vocal fold vibration is expected to be further extended to silent-speech monitoring and recognition.

Finally, the PNP offers extremely short response times, so it is suitable for vibration frequency detection. The PNP electronics was fixed to a tuning fork with an intrinsic frequency of 128 Hz (Fig. 5g). The tuning fork is excited by a knock and undergoes damped vibrations, which stimulate the PNP sensor to generate a correspondingly varying output voltage (Fig. 5h). This time-domain plot shows that the excitation intensity obviously affects the voltage amplitude. Also, based on the partial enlarged curves for 8 cycles (inset in Fig. 5h), it can be seen that the PNP detects the sinusoidal-like vibration modes of the tuning fork accurately. And from the cycle length, it can be calculated that the frequency is indeed 128 Hz. Next, we applied Fast Fourier Transform (FFT) to the original time-domain plot and obtained the corresponding frequency-domain plot (Fig. 5i). The main frequency signal of 128 Hz appears clearly in the frequency-domain plot, which once again proves the accuracy of vibration frequency detection by the PNP sensor. Similarly, the PNP is also able to precisely monitor the vibration signal of a 256 Hz tuning fork, and its schematic, time-domain and frequency-domain diagrams are shown in Figs. 5j–l in turn. Supplementary Video 4 also shows the detection of the tuning fork frequency by the PNP electronics. It is expected to utilize PNP film electronics as flexible vibration sensors to monitor the vibration status of different mechanical devices or to establish a haptic system for human-computer interaction.

## Discussion

In this study, we propose the concept of piezo-ionic-electric electronics, through which the all-polymer PNP film with highly-optimized electromechanical performance is presented. Hot-pressing provides thermal and pressure fields for obtaining the sandwich structure with broad and flat ion-electron interfaces. Benefiting from interfacial ionic-electronic effect, the PNP electronics acquires a particular piezo-ionic-electric working mechanism based on bend-self-polarization, while achieving efficient performance boost with $d_{33}$ up to -80.70 pC N$^{-1}$. The unique working mechanism is subdivided into two stages for elaboration and verification, centering on the interplay between the piezoelectric effect and the piezoionic effect. And the enhanced electrical output of the PNP is visually highlighted by comparison with the control groups. Moreover, this substantial enhancement is attributed to the large charge accumulation at the highly-polarized ion-electron interface synergized by the piezoionic effect. Testing has proven that the PNP electronics has more than one outstanding force-electric coupling parameter, including a pressure sensitivity of 51.50 mV kPa$^{-1}$, a response time of 9/11 ms, and a maximum peak power of 34.66 mW m$^{-2}$. Given these advantages, the PNP has been successfully utilized in energy harvesting, physiological signal monitoring, and vibration state detection, demonstrating the value of all-round piezo-generator applications. We have confidence that the paradigm of piezo-ionic-electric electronics as well as its intrinsic interfacial ionic-electronic effect will have a significant impact on the innovative development and valorized utilization of polymer-based, including PVDF-based and more non-fluoropolymer-based, piezo-generators.

## Methods

### Preparation of monolayer PVDF and Nafion film

PVDF powder (Kynar 720) was purchased from Arkema, and its density is 1.78 g cm$^{-3}$. PVDF was added to dimethylformamide (DMF) solvent. Then it was magnetically stirred in a 50 °C water bath for 2.5 h to obtain a uniform and transparent PVDF solution (0.15 g mL$^{-1}$). After removing air bubbles by vacuum oven, the solution was dripped onto a clean glass plate for blade-coating. The blade gap height is 500 μm and the moving speed is 15 mm s$^{-1}$. The glass plate was subsequently transferred to an oven at 60 °C and dried for 12 h, finally obtaining the monolayer coated-PVDF (CP). In addition, Nafion film (NR 211) was supplied by DuPont. After removing the backing film and coversheet with adhesive tape, the Nafion film can be used directly.

### Preparation of PVDF/Nafion/PVDF sandwiched multilayer film

The monolayer PVDF and Nafion films prepared above were stacked together as precursors in order. Then they were clamped in two homemade moulds for hot-pressing process. The films were pre-pressed at 140 °C for 10 min, then warmed up to 170 °C and pressed for 20 min, followed by water-cooled quenching under maintained pressure. Wherein, the pressure of hot-pressing was set at five gradient values from 100 MPa to 200 MPa. Finally, the desired PVDF/Nafion/PVDF multilayer film (PNP) were obtained at a pressure of 100 MPa. PVDF three-layer film (PPP) was prepared by stacking three separate monolayer films together and applying the same method as described above.

### Fabrication of flexible piezo-ionic-electric electronics

A dense Ag layer was deposited on both sides of the films as the electrode by the physical deposition method of radio-frequency magnetron sputtering. Subsequently, conductive carbon tape was adhered to the Ag electrodes (45 nm) on both sides as conducting wires to transfer the generated charges. Finally, they were sealed with polyurethane (PU) in order to block external interference. After the above procedure, flexible film devices of PNP, PPP and Nafion were produced. In addition, the PPP experienced an electrical-poling process prior to encapsulation, under a voltage of 2 kV for 10 min at ambient temperature.

### Materials characteristics

The cross-sectional morphology of the PNP was characterized using field emission scanning electron microscopy (FESEM, JSM-7800F, JEOL). The crystalline structure of PVDF was analyzed by X-ray diffraction (XRD, Empyrean, PANalytical) equipped with Cu/K$_\alpha$ radiation (wave-length is 0.1540 nm). Fourier transform infrared (FTIR, Nicolet iS50, Thermo Fisher) spectroscopy was used to characterize the conformational classes and relative contents belonging to the PVDF (in ATR mode), while the PNP constituents were confirmed (in transmission mode). Polymer films were thermally analyzed using differential scanning calorimetry (DSC, Discovery DSC 2500, TA) with a heating/

cooling rate of 5 °C min⁻¹. The morphology result based on the atomic force microscope was obtained by the Multimode 8 (Bruker). The probe (SCM-PIT-V2, Bruker) coated with Pt/Ir was applied for testing.

## Measurement of electrical performance

Linear motors (LinMot H01-23×86/160) and force gauges (M7-50, Mark-10) constitute an excitation source system with real-time display of pressure values. During device performance testing, the output voltage and charge signals were provided by Keithley 6514 system electrometer and output current signals were measured by a low-noise current amplifier (Stanford Research SR570). A low-noise voltage amplifier (Stanford Research SR560) was used for sensing signal detection during device application. All electrical signals were collected and analyzed by the Data Acquisition Card (NI PCI-6221).

## Human subject study

Two participants were recruited to participate in the device testing human pulse and vocal cord vibration experiments. Both participants were male and aged between 20-30 years old. The gender information is obtained based on the self-reporting method of the participants. Population characteristics are not relevant to the experiment or the results of this study. Informed consent from all participants was obtained before inclusion in this study. All human subject experiments were conducted by protocols approved by the Institutional Review Boards of Southwest Jiaotong University with the reference number of SWJTU-23012-NSFC(133).

## Reporting summary

Further information on research design is available in the Nature Portfolio Reporting Summary linked to this article.

## Data availability

The data that supports the findings of this study can be found in the Supplementary Information. Raw data are provided as a Source Data file. Source data are provided with this paper.

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

## Acknowledgements

This research is supported by the National Natural Science Foundation of China (No. 52303328), the Postdoctoral Innovation Talents Support Program (No. BX20220257), the Multiple Clean Energy Harvesting System (No. YYF20223026), the Sichuan Science and Technology Program (No. 2023NSFSC0313), and a Catalyst Seeding General Grant administered by the Royal Society of New Zealand (Contract 20-UOA-035-CSG). Thanks for the help from the Analysis and Testing Center of Southwest Jiaotong University.

## Author contributions

T.X., L.J., and W.Y. conceived the idea and designed the project. T.X. and L.J. carried out the experiment. T.X., L.J., Y.A., J.Z., and Y.S. analyzed the data and the mechanism. S.W., Y.Q., T.Y., and L.H. assisted in the experiment. T.X. and L.J. prepared the manuscript. W.Y., L.J., and W.D. funding acquisition. L.J. and W.Y. supervised the project. All authors participated in the analysis and discussion.

## Competing interests

The authors declare no competing interests.
