## [Transparent Peer Review file · Nature Communications]

All-polymer piezo-ionic-electric electronics

Corresponding Author: Professor Weiqing Yang

Version 0:

Reviewer comments:

Reviewer #1

(Remarks to the Author)

The authors have demonstrated an all-polymer piezo-ionic-electric electronics with PVDF/Nafion/PVDF (polyvinylidene difluoride) sandwich structure and regularized ion-electron interfaces. The devices shows an force-electric coupling enhancement with a d_{33} of ~ 80.70 pC N⁻¹, a pressure sensitivity of 51.50 mV kPa⁻¹ and a maximum peak power of 3.92 μ W.

Actually, this work reported by the authors have not shown new enough materials, devices, and working mechanisms. Polymer piezoelectric sensor, or even piezoionic sensor is common, piezo-ionic-electric electronics are not meaningful. The devices' performances are also common, and the demonstrations are routine. Accordingly, it is difficult to recommend this work to be published in high quality journal of Nature communications

Reviewer #2

(Remarks to the Author)

The manuscript entitled "All-polymer piezo-ionic-electric electronics" describes a piezo-generator system based on PVDF/Nafion/PVDF (PNP) sandwich structure that exhibits both piezoelectric and piezoionic effects. The proposed working principle of the presented piezo-ionic-electric electronics is based on piezoelectric activation by bend-self-polarization and piezoionic effect emanating from the Nafion due to pressure bending induced dissociation of protons within its nanochannels. The proposed concept is impressive in the field of piezo-generators, however, there are few inconsistent statements and unclear explanation of the working principle that need to be thoroughly clarified before publication.

1. The title "All-polymer piezo-ionic-electric electronics" seems to classify a novel piezo-system that exhibits both piezo-ionic and piezo-electric properties. The authors should properly define the theoretical principle that categorizes a device or system as piezo-ionic-electric for better evaluation of future systems based on this concept.

2. The authors claim that they have developed an "all-polymer" device based on PVDF/Nafion/PVDF sandwich structure. However, from Supplementary Fig.4, the electrodes for the prepared device were based on Ag electrodes sputtered on both sides of the PNP film. (i) The electrodes should be described as part of the piezo-ionic-electric device and not only PNP film. (ii) Ag is not a polymer, making the "all-polymer" claim partially inaccurate. For the terminology "all-polymer" to be true and accurate in the title, the electrode should be developed from a conductive polymer material such as PEDOT:PSS, Polyaniline, etc.

3. According to the authors, by simple repeated bending stimulation, PNP electronics can achieve self-polarized activation. However, self-polarized activation in PVDF based piezoelectric systems generally involves poling induced phenomena assisted by inorganic composite such as MXene (Nat. Commun. 12, 3171 (2021)). Please explain in detail how the self-polarized activation was achieved without high-voltage polarization or inorganic composite? Moreover, the "bend-self-polarization" should be explained in detail at the molecular level. The schematics presented in Fig. 1d, and Fig. 2 should display the polarization with respect to the poling of the β -phase structures. Also, the number of times repeated bending simulations were performed to achieve effective self-polarization should be specified.

4. The authors explained that bending stimulation induced the dissociation of the protons at the end of the side chains of the Nafion and directed movement in the nanochannels. While bending might alter the distribution or orientation of the sulfonic acid groups, it doesn't directly cause the dissociation of protons. Proton dissociation is primarily driven by chemical equilibrium (usually in an aqueous environment), not mechanical deformation.

5. The authors mentioned on page 8 line 165-167 that "Also, the vast majority of sulfonate groups in the Nafion intercalation has not yet dissociated, abstracting as positive ions (main chain and side chains with sulfonates) and negative ions (protons)". Generally, the sulfonic acid groups dissociate to release protons (positive ions) leaving behind negatively charged sulfonate groups. The protons are positive ions. Please double check your explanation.

6. In Fig. 2, the discussion on the dominance or decisive component between the two effects (piezoelectric or piezoionic) during the bending induced polarization is not clear. It was stated on page 10 that "Thus, it is the piezoionic effect that plays a decisive role in Stage II", whereas on page 11 it was stated that "Thus, it further justifies that the piezoelectric effect controls the piezoionic effect in Stage II". These two statements seem to contradict each other. (i) Please explain clearly the points at which each effect of the piezo-system plays a dominant role in the piezo-ionic-electric mechanism. (ii) Explain with supporting evidence how these two effects can be decoupled in the piezo-ionic-electric system.

7. Generally, piezo-ionic pressure sensors rely on pressure-induced EDL formation at the electrode/electrolyte interface for the sensing mechanism. In the piezo-ionic-electric working mechanism, is there a possibility of EDL formation at the ionic-electronic interfaces of the PNP film? And does it have any effect or contribution to the overall sensing performance of the device?

8. The authors mentioned on page 5 lines 109-111 that "Also, this hot-pressing method enables a stable combination of two structural basis, including the orientation alignment of the β -phase dipole and the directional movement of the inner protons nanophase channels". Please explain theoretically how the hot-pressing method enabled directional movement of the inner protons nanophase channels for better understanding.

9. There is no discussion on Fig. 4d, please double check and include the necessary explanation.

10. The application demonstrations in Figs. 5c-f do not effectively reflect the working principle of piezo-ionic-electric device. Most piezoionic pressure sensors and many other published works can perform these demonstrations making them less impactful in support of the working mechanism of this device.

Reviewer #3

(Remarks to the Author)

The manuscript entitled "All-polymer piezo-ionic-electric electronics" describes the formation of a sandwich structure of PVDF-Nafion-PVDF which undergoes self-poling via bending to produce a highlight pressure sensitive electromechanical device. The concepts within are well thought out, and tested appropriately. However, there are further methodology details which need to be provided by the authors prior to publication - particularly around the specific method used for piezoelectric testing. Once these methods are addressed appropriately, and the role of alternative charging processes (i.e. triboelectricity) ruled out - the manuscript would make a strong addition to Nature Communications, representing a simple, scalable approach to design high performance electromechanical devices. Please find my detailed comments below:

1. can the authors please provide the thickness of each component of the device in the main text and in Figure S4? Did the thickness of the nafion layer have an effect?

2. Did PPP undergo spontaneous polarisation from bending? If so - how did this occur as there is no electric field from the piezoionic effect to drive polarisation? If not - how is it comparable for testing?

3. the spontaneous polarization from bending is an extremely interesting result - as it would suggest the internal electric field created by the ionic effect is significant. Could the authors please show this process, i.e. electrical generation by bending over 1 to 10 cycles (or however many cycles are required for this process) - this will enable a significant benefit to future researchers. Was the short-circuit current in Figure 2 already showing this - if so, why didn't it increase with continued polarization?

4. Given the outstanding result of self-poling through bending, it raises the question of relative contribution between the piezoelectric polymer compared to the confined nafion itself? Is it possible to provide control experiments where nafion is sandwiched between two non-piezoelectric polymers (i.e. sheets of PTFE for a fluoropolymer comparison or another appropriate polymer) - again, this deconvolution of the magnitude of the effects would highlight the role of synergistic performance of piezo-ionic effects as opposed to simply ionic effects.

5. Can the authors please describe the piezoelectric testing methodology used in detail. The set-up described in Figure S6 appears to show that contact-separation mode is used, which induces significant triboelectrification in addition to piezoelectric and ionic effects? These methods need to report the force applied for testing, along with the waveform used - unless these methods are adequately described it is impossible to rule out alternative contributions to the charge generation mechanism (see: <https://onlinelibrary.wiley.com/doi/full/10.1002/aelm.202400019>; <https://www.nature.com/articles/s41467-022-29087-w>; <https://onlinelibrary.wiley.com/doi/abs/10.1002/adma.202002979>;)

6. The authors state force applied in Figure 4 ranged from 1 to 35N - were these from 0 to 1 up to 0 to 35N (i.e. contact-separation mode) or was a pre-load force used?

7. Can the authors please normalize their power and energy values to either per unit area or per device volume to enable comparison to prior works?
8. Did the two PVDF layers contact at high applied pressure? If so, the displacement of the liquid may explain the decreased performance above 220 kPa
9. In the introduction, can the authors describe the advantages of Piezoelectric-Ionic energy harvesters compared to piezoelectric-triboelectric energy harvesters (see: <https://doi.org/10.1021/acscami.1c04489>; <https://pubs.acs.org/doi/full/10.1021/acsaem.3c01196>) - these appear to be the direct comparative technology which could be produced via an air gap in the PVDF film.
10. While the work focusses on PVDF, the authors should highlight the potential of piezoelectric-ionic effects enable non-fluoropolymer-based energy harvesters (see: <https://doi.org/10.1002/sml.202311570>)

Version 1:

Reviewer comments:

Reviewer #2

(Remarks to the Author)

The authors have fully addressed all comments raised by us. Therefore, the revised manuscript can be published in the Nature Communications journal without any correction.

Reviewer #3

(Remarks to the Author)

The authors have performed significant new experiments and reworked their manuscript which has resulted in a dramatic improvement in the overall quality of the manuscript. In particular, the performance of experiments with PTFE-*nafion*-PTFE showing a negative result provides significant evidence in support of the claims within the manuscript - I thank the authors for taking the time to perform these experiments and produce such clear and significant changes.

The manuscript now contains significant novelty, clear methodology, and strong control experiments - and I believe the manuscript is suitable to be published in Nature Communications.

I have minor comments below:

- please add relevant references to Table S3: MXene (<https://doi.org/10.1038/s41467-021-23341-3>); CNT (<https://doi.org/10.1039/C9EE03059J>)

- the space charge of the structure demonstrated by COMSOL multiphysics Figure S19 should have a colour scale bar to denote the magnitude of the fields.

- the layer-by-layer space charge demonstrated in Figure S19 suggests the working mechanism of the device is similar to layered 3D TENGs (<https://onlinelibrary.wiley.com/doi/full/10.1002/aesr.202300259>), this doesn't need to be addressed here - but it's an interesting comparison.

List of Responses to the Reviewers' Comments

We would like to thank the reviewers for their valuable comments and suggestions on this manuscript, which greatly improves the quality of our manuscript. Following these comments and suggestions, we have made careful revisions to our previous manuscript and supplementary information (marked in red color), and provide response to the comments point-by-point as follows:

Reviewer #1 (Remarks to the Author):

The authors have demonstrated an all-polymer piezo-ionic-electric electronics with PVDF/Nafion/PVDF (polyvinylidene difluoride) sandwich structure and regularized ion-electron interfaces. The devices show a force-electric coupling enhancement with a d_{33} of $\sim 80.70 \text{ pC N}^{-1}$, a pressure sensitivity of $51.50 \text{ mV kPa}^{-1}$ and a maximum peak power of $3.92 \text{ }\mu\text{W}$.

Response:

We greatly appreciate the reviewers' valuable comments on our manuscript. And the response to these comments point by point is as follows:

Comment 1: Actually, this work reported by the authors have not shown new enough materials, devices, and working mechanisms.

Response:

As the reviewer comments, PVDF/Nafion/PVDF sandwich material and device structure is relatively common, which we likewise agree with. However, much unlike the common piezo-generators reports, our work is significantly innovative in piezo-material system, novel polarized interface, and unique device working mechanism. We chose a composite of two polymers, Nafion and PVDF, as the rare piezo-material. Nafion, a perfluorosulfonate ionomer, received very little attention in piezoionic materials. However, his structural (fluorocarbon

backbone) and functional (piezoionic effect) properties suggest that it is well suited to be combined with PVDF to develop the piezo-ionic-electric electronics, presented for the first time. Meanwhile, it is the simple sandwich structure that endows it with ion-electron interfaces, very special among numerous composite piezoelectric materials. The difference between the two free charge carriers and the high regularity make this ion-electron interface strongly polarized, which determines a large force-electric output. Further, the traditional piezoelectric effect and the emerging piezoionic effect interact at the interfaces, endowing the device with the piezo-ionic-electric working mechanism. It includes a bend-self-polarization as well as a coupled response, again with sufficient novelty.

Comment 2: Polymer piezoelectric sensor, or even piezoionic sensor is common, piezo-ionic-electric electronics are not meaningful.

Response:

We agree that the polymer piezoelectric sensor as well as the piezoionic sensor are more common today. But this does not mean that our proposed piezo-ionic-electric electronics is meaningless. On the one hand, benefiting from the interfacial ionic-electronic effect resulting from the interaction of above two piezo effects, such coupled electronics exhibits significant electromechanical performance enhancements. On the other hand, the piezo films of such devices have all-polymer properties. Unlike conventional organic-inorganic hybrid systems, all-organic systems can fundamentally avoid the trade-off between mechanical and electrical properties. In addition, as a simple piezoelectric activation method, its unique bend-self-polarization is expected to reduce the complication and risk of the common high-voltage polarization process. Thus, the piezo-ionic-electric electronics created by coupling the two effects derive several outstanding properties and interesting phenomena. It represents an innovative program for breaking through the highly optimized all-organic piezoelectric functional system, full of significance.

Comment 3: The devices' performances are also common, and the demonstrations are routine. Accordingly, it is difficult to recommend this work to be published in high quality journal of Nature communications.

Response:

As responded above, the ion-electron interfaces give the electronics remarkable force-electric coupling performance boost. Considering that the response mode of the novel device fundamentally belongs to piezoelectric mode, we follow the piezoelectric coefficient d_{33} as the key parameter of force-electric coupling performance, like other reports in the field. The test results show that the PNP film has a leading d_{33} of 80.70 pC N^{-1} . Here, to fully illustrate the performance advantages, we additionally selected several recent reports based on PVDF piezoelectric systems for comparison with our work. This part has been added as **Supplementary Fig. 17** and **Supplementary Table 3** (see below) in revised supplementary information.

Supplementary Fig. 17 Comparison of d_{33} values between PNP films and recent PVDF-based piezoelectric systems^{10,11,14-19}.

Supplementary Table 3

Comparison of d_{33} values between PNP films and recent PVDF-based piezoelectric systems.

Piezoelectric functional material	Self-polarizing or not	d_{33} (pC N ⁻¹)	References
PVDF/Nafion	Yes	80.7	Our work
PVDF/KNN	No	12	14
PVDF-TrFE	No	21	15
PVDF-TrFE/PC	No	49.1	16
PVDF	No	62	10
PVDF/CNTs	Yes	9.4	17
PVDF/Gly-MoS ₂	Yes	24.9	18
PVDF/BTO	Yes	51.2	11
PVDF/MXene	Yes	63.3	19

The related discussion has been added into the revised manuscript:

“In fact, as shown in **Supplementary Fig. 17** and **Supplementary Table 3**, the d_{33} of PNP is at the leading edge of PVDF-based piezoelectric systems, with or without additional electrical polarization.” (page 15)

Meanwhile, pressure sensitivity is likewise a classic metric for piezoelectric sensors. We have made a peer comparison in the **Fig. 1e** and **Supplementary Table 2**, which again illustrates the excellent performance of our novel device. In addition, we have already provided the PPP (hot-pressed PVDF three-layer film) device as a performance control group for longitudinal comparison in the supplementary information. Here we additionally added radar graphs of both (**Fig. R1**) to further demonstrate the performance optimization of our device over conventional piezoelectric devices.

Fig. R1 Radar graph comparing each key performance of PNP and PPP. where the response time is obtained from Supplementary Fig. 13 and the rest of the values were previously mentioned in the manuscript.

Reviewer #2 (Remarks to the Author):

The manuscript entitled “All-polymer piezo-ionic-electric electronics” describes a piezogenerator system based on PVDF/Nafion/PVDF (PNP) sandwich structure that exhibits both piezoelectric and piezoionic effects. The proposed working principle of the presented piezoionic-electric electronics is based on piezoelectric activation by bend-self-polarization and piezoionic effect emanating from the Nafion due to pressure bending induced dissociation of protons within its nanochannels. The proposed concept is impressive in the field of piezo-generators, however, there are few inconsistent statements and unclear explanation of the working principle that need to be thoroughly clarified before publication.

Response:

We are extremely grateful to the reviewer for recognizing our work and the proposed concept of piezo-ionic-electric electronics. All comments and concerns, especially on the working principle, were addressed point by point.

Comment 1: The title “All-polymer piezo-ionic-electric electronics” seems to classify a novel piezo-system that exhibits both piezo-ionic and piezo-electric properties. The authors should properly define the theoretical principle that categorizes a device or system as piezo-ionic-electric for better evaluation of future systems based on this concept.

Response:

Thanks to the reviewer for this constructive comment. The “piezo-ionic-electric electronics” does define a new type of piezo-system, which is different from mere piezoelectrics and piezoionics. Therefore, the theoretical principle for its classification are indeed crucial and forward-looking. At present stage, we interpret the piezo-ionic-electric electronics as “a self-powered piezo-generator that relies on both the piezoelectric effect and the piezoionic effect to realize the force-electric conversion function”.

In further detail, the piezo-ionic-electric electronics needs to meet at least three conditions:

- i) Piezo-material have at least two components for having carriers of both electron and ion, and thus both piezoelectric and piezoionic effects.
- ii) Interfaces (ion-electron interfaces) must be formed between the components rather than independently of each other.
- iii) The generated electrical signals from overall electronics are multisource but phase-consistent.

In the manuscript, we mainly use **Fig. 1a** to illustrate this new concept. In order to better elucidate this concept, we have revised the corresponding section “Concept, design and properties” in the manuscript. The following sentence has been added:

“In other words, it is defined as a self-powered piezo-generator that relies on both the piezoelectric effect and the piezoionic effect to realize the force-electric conversion function.”

(page 5)

Comment 2: The authors claim that they have developed an “all-polymer” device based on PVDF/Nafion/PVDF sandwich structure. However, from Supplementary Fig.4, the electrodes for the prepared device were based on Ag electrodes sputtered on both sides of the PNP film.

(i) The electrodes should be described as part of the piezo-ionic-electric device and not only PNP film. (ii) Ag is not a polymer, making the “all-polymer” claim partially inaccurate. For the terminology “all-polymer” to be true and accurate in the title, the electrode should be developed from a conductive polymer material such as PEDOT: PSS, Polyaniline, etc.

Response:

We appreciate the valuable comment from the reviewer. In fact, we use the term “all-polymer” to highlight the all-organic characteristics of PNP films, which is distinctive among all the existing piezoelectric composites. Unlike conventional organic-inorganic hybrid systems, all-

polymer systems can fundamentally avoid the trade-off between mechanical and electrical properties, which is a notable trend in piezo-electronics. At the same time, the electrodes in the piezo-ionic-electric electronics only act only as conductors, and their type has almost no effect on the fundamental laws and performance. This is different from the role of electrodes in supercapacitors, optoelectronic devices, etc. Therefore, placing “all-polymer” as a modifier before “piezo-ionic-electric” will highlight the key all-polymer characteristic of the piezo-ionic-electric system in a concise manner, while making the title sufficiently attractive.

Following the reviewer’s great suggestion, we replaced the Ag electrodes with PEDOT:PSS electrodes, Polyaniline electrodes, as well as the classical Copper (Cu) electrodes. The piezoelectric test results show that the electrode type has little effect on the performance of the PNP device, as shown in **Fig. R2** (see below).

We dropped a DMF (dimethylformamide) solution of PEDOT:PSS onto the surface of the PNP film to prepare the device with the structure shown in **Fig. R2a**. The butanol ($C_4H_{10}O$) dispersion of polyaniline (PANI) was drop-coated on surface of the PNP film to obtain the device in **Fig. R2b**. The device in **Fig. R2c** was obtained by attaching copper foils tightly to both sides of the PNP film. These electrodes are all circles with 6 mm radius, which is consistent with the Ag electrodes in the manuscript. Under a pressure of 44.23 kPa, above three devices (after bend-self- polarization) were tested for open-circuit voltage (**Figs. R2d-f**) and short-circuit current (**Figs. R2g-i**). The output magnitude was essentially the same as that of the original PNP with Ag electrodes at the same pressure (voltage about 3 V, current about 150 nA, **Figs. 4a** and **b**). Therefore, whether metal electrode or polymer electrode, its effect on the performance of the piezo-ionic-electric device is negligible. In other words, if the original Ag electrode is replaced by polymer electrode, the physical laws and basic properties of the electronics will not change. The electrode is not the key to the piezo-ionic-electric system.

In this way, we wish to clarify the intent and significance of this title.

Fig. R2 Schematic structure of a PNP electronics with **a** PEDOT:PSS electrode, **b** PANI electrode, and **c** Cu electrode, respectively. Under 44.23 kPa pressure, the open-circuit voltage curves of **d** PNP device with PEDOT:PSS electrodes, **e** PNP device with PANI electrodes, and **f** PNP device with Cu electrodes. Under 44.23 kPa pressure, the short-circuit current curves of **g** PNP device with PEDOT:PSS electrodes, **h** PNP device with PANI electrodes, and **i** PNP device with Cu electrodes. The force-electrical coupling outputs are all essentially consistent with the original device with Ag electrodes.

Comment 3: According to the authors, by simple repeated bending stimulation, PNP electronics can achieve self-polarized activation. However, self-polarized activation in PVDF based piezoelectric systems generally involves poling induced phenomena assisted by inorganic composite such as MXene (Nat. Commun. 12, 3171 (2021)). Please explain in detail how the self-polarized activation was achieved without high-voltage polarization or inorganic

composite? Moreover, the “bend-self-polarization” should be explained in detail at the molecular level. The schematics presented in Fig. 1d, and Fig. 2 should display the polarization with respect to the poling of the β -phase structures. Also, the number of times repeated bending simulations were performed to achieve effective self-polarization should be specified.

Response:

We thank the reviewer for these professional comments and the interest in the mechanism of bend-self-polarization. In recent years, self-polarization phenomena in PVDF-based piezoelectric systems have often originated from the polarization induction of polar fillers, such as MXene (Nature Communications 12, 3171 (2021)), glycine-modified molybdenum disulfide (Gly-MoS₂, Nano Energy 99, 107379 (2022)), and (Pb, Zr)TiO₃ (PZT, Nano Energy 98, 107340 (2022)). These nanomaterials have strong electrostatic interactions with PVDF, inducing the polymer to achieve effective local polarization locking.

In contrast to these principles, the polarization in PNP devices originates from the piezoionic ion-based electric field generated by the sandwiched Nafion under bending conditions. The same direction and repeated bending stimulus then generate an AC-like polarized electric field. It induces the upper and lower PVDF dipoles to align in the same direction, and acquire spontaneous polarization (P_s). After bending behavior stops, the PVDF chains partially relax from P_s to residual polarization (P_r), thus achieving an overall polarization. A key point is the occurrence of ion-electron interactions at the special ion-electron interfaces, which amplifies interface-based polarized electric field and enhances the ability to polarize the PVDF.

Figs 1d and 2a(i-iii) visualize this process. As suggested by the reviewer, we have added the polarization of the β -phase structure of PVDF in these two figures (**arrows and annotations**):

“Fig. 1 | Concept, design and properties of the piezo-ionic-electric electronics. d After repeated bending stimulation, the PNP electronics achieves piezoelectric activation, a particular process called bend-self-polarization.”

“Fig. 2 | The unique piezo-ionic-electric working mechanism of the PNP electronics and its verification. a Schematic diagram of the two stages (seven elements in total) in the working process of the PNP electronics, abstractly depicting the microstructural evolution inside the PNP.....”

In order to explain the bending self-polarization in more detail at the molecular level, we have optimized the original discussion about bend-self-polarization in the revised manuscript:

“Element a(i) shows the initial state of the PNP electronics obtained by laminated hot-pressing. The β -phase within the upper and lower PVDF layers is abstracted as dipoles disorderedly arranged in the out-of-plane direction (polarization $P \approx 0$), even though most of them tend to be oriented under the constraint of the pressure and thermal fields. Also, the sulfonate groups and free protons of Nafion interlayer are in charge equilibrium, abstracting as negative ions (main chain and side chains with sulfonates) and positive ions (protons). Subsequently, the bending stimulation of the film electronics induces drives the dissociated protons to move directionally in the nanochannels toward the in-plane expansion side, which stems from the piezoionic effect of Nafion. Repeated bending stimulations with the same bending moment direction induces an ion-induced electric field in the intercalation. In turn, the built-in electric field leads to intense ion-electron interactions between the two ion-electron interfaces. It amplifies the polarization induction, promoting the internal alignment and overall orientation (spontaneous polarization $P_s > 0$) between the dipoles of the two PVDF layers, as depicted in Element a(ii). Finally, Element a(iii) shows the completion of the self-polarization process: the ions at the intercalation are uniformly distributed but more easily migrated, while the upper and lower dipoles achieve consistent polarization alignment after molecular chains relaxation (residual polarization $P_r > 0$).” (page 8-9)

Finally, following the great advice from the reviewer, we have explored the bending cycles for achieving effective self-polarization through a large number of PNP samples. The corresponding test procedure and results are shown in **Supplementary Fig. 6** and have been added to the revised supplementary information.

Supplementary Fig. 6a is a diagram of the experimental setup for applying the bending stimulus to PNP devices. We flatten the PNP film device on two acrylic holders. The back and forth motion of the linear motor can apply the bending stimulation to the film at different degrees (curvatures). In order to accurately quantify the degree of bending behavior in terms of curvature, we utilized the film bending model for mechanical analysis, as shown in

Supplementary Fig. 6b. The PNP film is tightly centered on the PU encapsulation layer (only one side is shown). When the ends are compressed by a distance ΔL , the bending of the PU will cause the PNP film to bend with the same curvature. Their out-of-plane displacement (h) can be calculated as $h = A(1 + \cos(2\pi x_1/L))/2$, where A is the bending amplitude and L is the initial length of the film device (The length excluding the fixed portion at each end). Considering that the PNP film is at the center of the PU and its length is significantly smaller than the PU, the curvature (ω) of the PNP film can be calculated at $x_1 = 0$ (at the center of the device). It is given as $\omega = (-4\pi\sqrt{\Delta L/L})/L$ (Proceedings of the National Academy of Sciences of the United States of America 111, 1927-1932 (2014)). Accordingly, the curvature ω of a PNP film at a certain degree of bending can be calculated from the compression distance ΔL .

In order to quantify the bending cycle for bending self-polarization, we fix the bending frequency ($f = 1$ Hz) and count the number of bending at three typical compression distances ($\Delta L_1 = 10$ mm, $\Delta L_2 = 20$ mm, $\Delta L_3 = 30$ mm). **Supplementary Fig. 6c** show the actual photographs of the three states. Their curvatures (taking absolute values without considering directionality) increase with the compression distance as $\omega_1 = 0.192$ mm⁻¹ , $\omega_2 = 0.271$ mm⁻¹ and $\omega_3 = 0.332$ mm⁻¹, respectively. A group of 20 device-samples under each bending curvature was tested for number of bending. The statistical results are depicted in box-whisker plots and distribution curves, as shown in **Supplementary Fig. 6d**. The average number of bending for the three curvatures was 278, 125 and 31.75 times, respectively. It is clear that there is a negative correlation between the number of bending and the bending curvature. In fact, greater curvature implies a larger gradient of in-plane strain, which in turn induces a larger piezoionic electric field (Chemical Engineering Journal 482, 148988 (2024); Nano Letters 21, 5369-5376 (2021); Small 12, 5074-5080 (2016)). As a result, the polarization effect is subsequently more pronounced and the number of bends required to achieve the polarized state is decreased. Our experimental data is consistent with this theory. On the other

hand, the standard deviation of the number of bending is greater at small curvatures. This is due to the fact that the experimental error becomes larger as the number of bends increases. By now, the number of bending in the three typical bending states is basically clear, adding important regularities to the development of the piezo-ionic-electric system.

Supplementary Fig. 6 Experiments about bending number for achieving bend-self-polarization. **a** Diagram of the experimental setup for applying the bending stimulus to PNP devices. **b** Bending mechanical model of PNP film-device. **c** Actual photographs of the three typical bending states the corresponding curvatures. **d** Statistical plot of the number of bending required to achieve polarization activation under three curvature conditions, where the statistical distribution of the number of bending is depicted by curves. Boxes and whiskers are drawn as the 25-75th and 5-95th percentiles, respectively. The inset shows the mean values of the number of bending and their relationship curves under three bending curvatures, and the overlapping part indicates the standard deviation.

A brief discussion to **Supplementary Fig.6** was also added to the revised manuscript:

“Moreover, a large number of samples have shown statistically that the number of bending required to achieve polarization exhibits a negative correlation with the bending curvature (Supplementary Fig. 6, details in Supplementary Note 1). At approximate limiting curvature (0.332 mm⁻¹), the average number of bending is about 32 times.” (page 8)

In addition, a more detailed discussion of the bending number experiment is added to the **Supplementary Note 1** in the revised supplementary information.

Comment 4: The authors explained that bending stimulation induced the dissociation of the protons at the end of the side chains of the Nafion and directed movement in the nanochannels. While bending might alter the distribution or orientation of the sulfonic acid groups, it doesn't directly cause the dissociation of protons. Proton dissociation is primarily driven by chemical equilibrium (usually in an aqueous environment), not mechanical deformation.

Response:

Thank the reviewer for the professional comment. We totally agree that mechanical stimulation of bending cannot directly induce dissociation of protons in the sulfonic acid. We apologize for our previous misunderstanding, which has been corrected now:

In hydrated Nafion, such as the Nafion film (NR 211, from DuPont) used in this work, protons exist as hydronium ion (Science 385, 1115-1120 (2024); International Journal of Hydrogen Energy 44, 28919-28938 (2019)). Therefore, the proton dissociation occurs during the preparation process. Upon subsequent mechanical stimulation, dissociated protons move within the nanochannel toward towards the in-plane expansion side, without further proton dissociation process occurring. This proton transport process follows the proton hopping mechanism (Grotthuss) and the matrix transport mechanism (vehicular) (Journal of Physical

Chemistry C 122, 9710-9717 (2018)).

Following this correct view, we have revised several relevant discussions in the manuscript, as follow:

- i) “Stimulated by non-uniform stress, **ionomer Nafion undergoes non-uniform internal ion migration**, which in turn generates ionic potential differences, referred to the piezoionic effect” (page 3-4)
- ii) “Also, **the sulfonate groups and free protons of Nafion interlayer are in charge equilibrium**, abstracting as negative ions (main chain and side chains with sulfonates) and positive ions (protons). Subsequently, **the bending stimulation of the film electronics induces drives the dissociated protons to move directionally in the nanochannels toward the in-plane expansion side**, which stems from the piezoionic effect of Nafion.” (page 9)
- iii) “Similarly, **pure Nafion ion-device (after repeated bending stimulation) is able to rely on the migration of protons dissociated from sulfonate groups in the nanoscale ion channels to realize piezoionic effects under stress**, generating an ionic-induced electric field, and thus also outputting electrical signals” (page 16)

Comment 5: The authors mentioned on page 8 line 165-167 that “Also, the vast majority of sulfonate groups in the Nafion intercalation has not yet dissociated, abstracting as positive ions (main chain and side chains with sulfonates) and negative ions (protons)”. Generally, the sulfonic acid groups dissociate to release protons (positive ions) leaving behind negatively charged sulfonate groups. The protons are positive ions. Please double check your explanation.

Response:

Many thanks to the reviewers for checking out this mistake of principle. We apologize for our clerical error. We have corrected it in the revised manuscript:

“....., abstracting as **negative** ions (main chain and side chains with sulfonates) and **positive**

ions (protons)” (page 9)

Again, we apologize for the oversight. In order to avoid such errors still existing, all authors have scrutinized the manuscript.

Comment 6: In Fig. 2, the discussion on the dominance or decisive component between the two effects (piezoelectric or piezoionic) during the bending induced polarization is not clear. It was stated on page 10 that “Thus, it is the piezoionic effect that plays a decisive role in Stage II”, whereas on page 11 it was stated that “Thus, it further justifies that the piezoelectric effect controls the piezoionic effect in Stage II”. These two statements seem to contradict each other. (i) Please explain clearly the points at which each effect of the piezo-system plays a dominant role in the piezo-ionic-electric mechanism. (ii) Explain with supporting evidence how these two effects can be decoupled in the piezo-ionic-electric system.

Response:

We appreciate the reviewer’s valuable comments. According to the analytical results of **Fig. 2b** and **Fig. 2d**, it is Stage I, not Stage II, that is decisively controlled by the piezoionic effect. We apologize for the mistake in writing and the contradictory statements here. In the revised manuscript, the sentence “Thus, it is the piezoionic effect that plays a decisive role in Stage II” on page 11 had been changed to “**Thus, it is the piezoionic effect that plays a decisive role in Stage I**”. And the original expression “Thus, it further justifies that the piezoelectric effect controls the piezoionic effect in Stage II” on page 11 is correct.

The different roles played by the two effects in the unique piezo-ionic-electric mechanism are explained in further detail:

i) Briefly, Stage I is the piezoionic electric field from Nafion interlayer under bending stimulation, which in turn induces dipole orientation and alignment of the upper and lower

PVDF layers. This process is called bend-self-polarization, attributed to strong ion-electron interactions at the interface. The **Fig 2a(i-iii)** and **Fig R2a** (i.e., **Fig 1d**) depicts the process. Apparently, the source of this polarization is the ion-based electric field generated by the piezoionic effect of Nafion. Thus, in the manuscript we emphasize that “it is the piezoionic effect that plays a decisive role in Stage I”.

And in Stage II, the film is pressurized to undergo out-of-plane strain, inducing the upper and lower layers of PVDF produce dipole changes in the same direction. Subsequently, Nafion undergoes ionic movement under the confinement of the upper and lower electric fields, producing an overall ion-electron dual-polarization. The following dynamic response process is similar to the conventional piezoelectric response. The **Fig 2a(iv-vii)** and **Fig R2b** (added as **Supplementary Fig. 5** in the revised supplementary information) describes the process. Therefore, the piezoelectric effect of the upper and lower PVDF layers, as a direct response part of the out-of-plane strain, controls the piezoionic effect and the overall signaling.

Fig. R3 Brief logic of the two stages in the unique piezo-ionic-electric working mechanism. **a** The ion-based built-in electric field from the piezoionic effect induces piezoelectric polarization of the PVDF layers. **b** The piezoelectric effect responds to out-of-plane strains and constrains the piezoionic movement, resulting in an overall signal.

ii) In order to validate the mechanism and further clarify the different roles of the two effects

in the two stages, we designed the corresponding validation experiments in the original manuscript. By exploring the correspondence between the initial bending direction and the piezoelectric phase after activation, the dominant role of the piezoionic effect in Stage I is confirmed. In **Figs. 2b** and **d**, two unactivated PNP devices, each stimulated by bending moments in different directions (+Z and -Z), produce opposite current phases when stressed. Considering that the two devices are wired in exactly the same way in the piezoelectric test system, this difference can only come from a difference in the activation conditions (i.e., bending direction). Obviously, bending moments of different directions induce opposite directions of ion motion, which in turn produce opposite ion-based electric fields in the out-of-plane direction. Therefore, the overall polarization direction of the PVDF in the two devices is consequently opposite, which in turn generates short-circuit currents with opposite phases. This supports the decisive role of the piezoionic effect in Stage I.

On the other hand, for a PNP device that has been activated by bending in a certain direction and then bending it approximately in the reverse direction, there is no change in the phase of the current when it is pressed, as shown in **Figs. 2c** and **e**. This demonstrates the fact that dipoles of PVDF layers, which has been oriented by bending, is difficult to be reversed again by the opposite built-in ionic electric field, since such a reversal often requires a huge external directed energy. In other words, the direction of polarization of the bent self-polarized PVDF is relatively stable and is no longer easily affected by the ion-based electric field. Therefore, in Stage II, the piezoelectric effect responds to the pressure stably, constraining the piezoelectric ion effect and realizing the overall force-electric response.

In order to directly illustrate the significance of the above two sets of comparison experiments, we have modified the legend of **Figs 2b-e**, as follow:

“Fig. 2 | The unique piezo-ionic-electric working mechanism of the PNP electronics and its verification. **b** The process and principles of the validation experiments in Stage I. When the two PNPs were bent in opposite directions (-Z and +Z), the difference of internal microscopic evolution are compared schematically. **c** The process and principles of the validation experiments in Stage II. For the same PNP, bending in one direction (-Z) was followed by bending in the other direction (+Z), whose micro-evolution is also shown. **d** Plot of opposite short-circuit current phases, supporting the decisive role of the piezoionic effect in the bend-self-polarization. **e** Plot of unchanged short-circuit current phase, reflecting the leading role of piezoelectric effect in the overall response process.”

In addition to the above original validation experiments, here we supplemented the short-circuit phase calibration experiments to further confirm the role of the two effects in the mechanism. This relevant process has been added as **Supplementary Fig. 10** in revised supplementary information. As shown in **Supplementary Fig. 10a**, we connect the signal generator directly to the electrometer, which is equivalent to an AC power supply directly connected to an ammeter. Keep the positive pole of the power connected to the red wire of the ammeter and the negative

terminal connected to the black wire, for subsequent experimental control. The signal generator produces a square wave signal with a period of 1s, a duty cycle of 50%, and high and low levels of 1 V and 0 V as an AC source (**Supplementary Fig. 10b**). The open-circuit current measured by the electrometer was -0.2 mA and 0 mA at high and low levels, respectively (**Supplementary Fig. 10c**). Thus, it is concluded as follows: when the positive pole of the power is connected to the red wire of the electrometer, the value of the short-circuit current measured is negative.

Based on this important law, the equivalent positive and negative poles of a PNP device when pressed or released can be accurately determined. Undoubtedly, this helps to further justify bend-self-polarization. Thus, in **Supplementary Fig. 10d**, when a bending moment in the +Z direction is applied to the PNP, the free protons in the Nafion move toward the top (surface under tension) (Science 376, 502-507 (2022)). The formed piezoionic electric field is from the top to the bottom, which is also the direction of the PVDF dipole after self-polarization. Therefore, when the device is pressurized, its top electrode is equivalently a positive pole and the bottom is a negative pole. Under the same wiring with the calibration circuit, it is supposed to produce a negative current value. The experimental results in **Supplementary Fig. 10e** are consistent with this speculation, with the device producing a downward short-circuit current when pressed. Thus, it is indeed the piezoionic effect under the bending stimulus that aligns the dipole of the PVDF in Stage I. In the subsequent Stage II, the activated PVDF first makes a piezoelectric response, constraining the piezoionic effect, which together produce an overall electrical signal that conforms to the regularity.

Supplementary Fig. 10 Current phase calibration to assist in verification of bend-self-polarization dominated by piezoionic effect. **a** Electrometer synergized with signal generator for calibrating the short-circuit current phase, and the corresponding equivalent circuit. **b** The signal generator produces a square wave signal as a power source. **c** The square-wave short-circuit current detected by the electrometer is negative value. **d** After bending the PNP device in the +Z direction, according to the principle of bend-self-polarization, the tensile surface is equivalent to the positive pole of the power supply, and the compressed surface is the negative pole (when pressed). **e** With the wire connections consistent with calibration circuit, the device produces a negative current when stressed, which is in accordance with the results of the calibration circuit.

At present, the roles played by the two effects in the piezo-ionic-electric mechanism have been largely clarified and decoupled. Discussion of **Supplementary Fig. 10** was also added to the revised manuscript:

“In addition, we supplemented the short-circuit phase calibration experiments (Supplementary Fig.10 and Supplementary Note 3 for details). According to the experimental results, for the PNP device, the tensile side during bend activation can be equated to the positive pole of the power (when pressurized). This is consistent with Nation’s theory of piezoionic effect²⁶. It further validates the piezo-ionic-electric mechanism and the respective roles of these two effects in it.” (page 11)

Meanwhile, a more detailed discussion is likewise added to the **Supplementary Note 3** in revised supplementary information.

Comment 7: Generally, piezo-ionic pressure sensors rely on pressure-induced EDL formation at the electrode/electrolyte interface for the sensing mechanism. In the piezo-ionic-electric working mechanism, is there a possibility of EDL formation at the ionic-electronic interfaces of the PNP film? And does it have any effect or contribution to the overall sensing performance of the device?

Response:

We thank the reviewers for the constructive comments. The electrical double layer is a well-known concept and principle in supercapacitors filed. In recent years, as a new type of capacitive sensing, the iontronic sensing is precisely built on the EDL layer formed by ionic electronic contact (Advanced Materials 33, 2003464 (2021)). Interestingly, we similarly speculated that an EDL structure was formed in the piezo-ionic-electric working mechanism early in this study. But as the research progressed, we did not take the concept of EDL to describe the interface in this work, for the following reason:

In a narrow sense, piezoionic sensors are force-sensitive devices that generate an ion concentration gradient under pressure gradient. Clearly, the piezoionic effect belongs to the self-powered pressure sensing, fundamentally different from the iontronic sensing with passive characteristics. This difference is also reflected in the interface phenomena. The EDL in iontronic sensors comes from the electrode/ionic-material interface, while our proposed ion-electron interface originates between the dielectric and the ionic material. This structural difference determines the completely different working modes. In iontronic sensing, an external constant power supply is necessary for the formation of the EDL and for its capacitance to change under pressure. However, the ion-electron interface in our work derives only from the interfacial polarization under pressure, which serves to transmit the electric field and enhance the output. It is not a sensing functional unit like EDL. The true sensing dependent variable of piezo-ionic-electric electronics is the overall polarizability of the material.

Therefore, to avoid misunderstanding, the ion-electron interface in this work cannot be equated with the EDL in iontronic sensing.

Comment 8: The authors mentioned on page 5 lines 109-111 that “Also, this hot-pressing method enables a stable combination of two structural basis, including the orientation alignment of the β -phase dipole and the directional movement of the inner protons nanophase channels”. Please explain theoretically how the hot-pressing method enabled directional movement of the inner protons nanophase channels for better understanding.

Response:

We appreciate the reviewer this value comment. In fact, the expression of this statement was ambiguous, for which we sincerely apologize. We have provided a detailed explanation and corrected the statement, as follows:

What we originally intended to convey was that hot-pressing process can stabilize the two

structural foundations together to jointly realize the force-electric conversion function. Also, hot-pressing method does not induce directional movement of protons and their nano-channels. This nanophase channel is an intrinsic structural property of Nafion, a typical nanophase separated polymer (ACS Nano 16, 19240-19252 (2022)). As well, the directional movement of protons occurs when they are subjected to non-uniform external forces, which is also not realized by hot-pressing. Thus, the structural basis of the Nafion piezoionic effect is spontaneous during other molding processes, independent of the hot-pressing method.

In this work, hot-pressing is a key process for the preparation of PNP films, which can provide a directed pressure field and a precise thermal field. On the one hand, the hot pressing can induce PVDF chain orientation to generate the β -phase through melt state energy injection (Nature Communications 10, 4535 (2019); Nature Communications 15, 819 (2024)). This is an important structural basis for piezoelectric polarization in the piezo-ionic-electric electronics. On the other hand, the high temperature and pressure can induce the chain segments of two polymer molecules, Nafion and PVDF, to diffuse with each other and form a stable and ordered ion-electron interface. Therefore, being hot-pressed is essential for the piezo-ionic-electric device to realize its basic functions.

In order to completely correct the unclear formulation, we have changed this sentence “Also,nanophase channels” in the revised manuscript as follow:

“At the same time, the critical structures on both sides of the interface, including the β -phase effective for piezoelectric effect and the intrinsic protons nano-channels for piezoionic effect, are bound into a functional unity.” (page 6)

Comment 9: There is no discussion on Fig. 4d, please double check and include the necessary explanation.

Response:

Thanks for the reviewer's careful check. We apologize for our errors and omissions in the original manuscript. Incorrect figure matching occurs at this and nearby locations. We have revised **Fig. 4d** and **Fig. 4e** at the correct discussion sentence:

“And Fig. 4d illustrates the voltage signal profile in a working cycle (17.5 N, 5 Hz), from which it can be obtained that the PNP has a remarkably fast pressure response time, T_r/T_f of 9/11 ms. Then, the magnitude of voltage output at different frequencies (1 Hz, 2 Hz and 5 Hz) is compared, reflecting its good frequency stability (Fig. 4e).” (page 20)

Sorry again. In revising, we once again scrutinized the entire manuscript.

Comment 10: The application demonstrations in Figs, 5c-f do not effectively reflect the working principle of piezo-ionic-electric device. Most piezoionic pressure sensors and many other published works can perform these demonstrations making them less impactful in support of the working mechanism of this device.

Response:

Thanks to the reviewer for the constructive comment. The main intention of the various applications in **Fig 5** is to demonstrate the all-round excellent performance of the piezo-ionic-electric device and their suitability for various application scenarios, rather than to provide additional application supports for the piezo-ionic electric working mechanism.

Firstly, **Figs 5c-f** demonstrate the accurate detection of weak signals from the human body, including pulse waves and vocal cord vibrations. This undoubtedly reflects the high pressure

sensitivity of the device. Although many works have similar applications, our high-performance device is able to accurately test the distinct syllable peaks of a single word. In particular, each characteristic peak of the pulse within a single cycle is shown perfectly, which is often crippledly displayed in other work. Thus, these application demonstrations highlight the outstanding sensing capabilities and application performance of our piezo-ionic-electric devices.

Next, despite the specificity of this piezo-ionic-electric mechanism, the devices are consistent with conventional piezoelectric film devices in terms of their operating modes. Both are subjected to vertical pressure to generate an electrical signal, so the application areas largely overlap. The difference is that the piezo-ionic-electric device performs better in the same application. For example, although the PNP is only 70 μm thick, it can light up 26 small LEDs in series when pressed by the thumb. Meanwhile, the flexible device can be attached to irregular surfaces (angled tuning forks) to accurately detect vibration frequencies of up to 256 Hz. These two examples highlight our device's high output power and extremely fast response time, respectively. Therefore, it is difficult to support the working mechanism with the help of these application demonstrations. Even so, our devices tend to perform better.

Reviewer #3 (Remarks to the Author):

The manuscript entitled “All-polymer piezo-ionic-electric electronics” describes the formation of a sandwich structure of PVDF-Nafion-PVDF which undergoes self-poling via bending to produce a highlight pressure sensitive electromechanical device. The concepts within are well thought out, and tested appropriately. However, there are further methodology details which need to be provided by the authors prior to publication -particularly around the specific method used for piezoelectric testing. Once these methods are addressed appropriately, and the role of alternative charging processes (i.e. triboelectricity) ruled out - the manuscript would make a strong addition to Nature Communications, representing a simple, scalable approach to design high performance electromechanical devices. Please find my detailed comments below:

Response:

We greatly appreciate the reviewer’s affirmation of the concepts, methods, and implications of this work, which strengthens our confidence to continue exploring this innovative system. All concerns and comments, especially specific methods used for piezoelectric testing, were addressed and responded point by point.

Comment 1: Can the authors please provide the thickness of each component of the device in the main text and in Figure S4? Did the thickness of the nafion layer have an effect?

Response:

Thank the reviewer for this constructive comment. We have tested the thickness of each layer of the PNP device by several methods. The test results were supplemented in **Supplementary Fig. 4** (as below). We tested the morphology at the step area between the PNP film and the sputtered Ag electrode by atomic force microscopy (AFM) (**Supplementary Fig. 4b**). The height distribution of surface on the linear path between position A and position B is shown in **Supplementary Fig. 4c**. From this, the thickness of the Ag electrode can be obtained as 45 nm. The thickness of the encapsulation layer polyurethane (PU) was obtained by a thickness gauge,

and the result was $46\mu\text{m}$ (**Supplementary Fig. 4d**). In addition, we analyzed the cross-section SEM image of the PNP and obtained the thicknesses of the upper and lower PVDF as well as the Nafion interlayer as $19\mu\text{m}$, $9\mu\text{m}$, and $22\mu\text{m}$, respectively (**Supplementary Fig. 4e**).

Supplementary Fig. 4 Layer structure of the PNP flexible piezo-ionic-electric device and thickness test of each layer. **a** Schematic representation of the layer components of the PNP device. **b** Atomic force microscope (AFM) morphology result at the step between PNP film and Ag electrode edge. The scale bar is $5\mu\text{m}$. **c** Distribution of surface height along the straight line from position A to position B. The illustration shows the step from PNP film to Ag electrode. **d** Thickness of the encapsulation layer PU form a thickness gauge. **e** Thickness of layers in PNP film obtained from cross-section SEM image. The PVDF layers are orange areas, the Nafion interlayer is blue.

Also, for visual presentation, we have summarized the thickness of each layer in Supplementary Table 1, added to the revised supplementary information:

Supplementary Table 1

Thickness of constituent layers in PNP thin film devices.

Layer part	PU	Ag	PVDF (upper)	Nafion	PVDF (below)
Thickness	46 μm	45 nm	19 μm	9 μm	22 μm

Correspondingly, we improved the discussion of the layered structure of the PNP device in the revised manuscript:

“Supplementary Fig. 4 and Supplementary Table 1 present the final structure of the prepared flexible PNP piezo-ionic-electric electronics as well as the testing results on the thickness of each layer.” (page 6)

On the other hand, we consider that the thickness of the Nafion layer is not a critical factor affecting the performance of the PNP device. For the sandwiched layer Nafion, it is the intensity of its piezoionic effect that affects the overall force-electric coupling output. On this basis, increasing the proton mobility by means such as increasing the concentration of protons helps to achieve this. For example, recent works have demonstrated that amplifying the difference in anion and cation mobility helps to enhance the piezoionic effect (Advanced Materials 36, 2313127 (2024); Advanced Materials 36, 2307875 (2024)). Therefore, a moderate change in the thickness of Nafion will not affect the overall performance of the PNP electronics.

Comment 2: Did PPP undergo spontaneous polarisation from bending? If so - how did this occur as there is no electric field from the piezoionic effect to drive polarisation? If not - how is it comparable for testing?

Response:

We thank the reviewer for this comment, beneficial to improve our work. PPP does not possess the bend-self-polarization property, but requires a conventional electric-polarization process to excite its piezoelectricity. We apologize for not explicitly indicating this in the original manuscript. The relevant explanations and additional modifications are provided below:

PPP means PVDF-PVDF-PVDF, which is obtained by stacking three independent PVDF films and then hot-pressing them. Without the piezoionic electric field from Nafion inside the PNP, the PPP cannot achieve self-polarization from the bending. Therefore, conventional electrical-poling process is used for PPP activation. In detail, The PPP film (has been sputtered with Ag electrodes) is placed in two polar plates and DC high-voltage (2 kV) is applied for certain time (10 min) at room temperature. The corresponding diagram was added as **Supplementary Fig. 12** (as below) in the revised supplementary information. Such polarized PPP film device acquires piezoelectric properties, available for testing and comparison. The significance of PPP as a control group is to highlight two major implications of Nafion interlayer in this work: the unique self-polarization mechanism and the polarization-enhanced output by the ion-electron interface.

Supplementary Fig. 12 Schematic diagram of electrical-poling for PPP film. The applied voltage was a DC high voltage of 2 kV for 10 min.

Corresponding changes have been added in the revised manuscript:

- i) “Note that PPP experienced an electrical-poling process beforehand, as detailed in the experimental section and in the Supplementary Fig. 12.” (page 14)
- ii) “In addition, the PPP experienced an electrical-poling process prior to encapsulation, under a voltage of 2 kV for 10 min at ambient temperature.” (page 27)

Comment 3: The spontaneous polarization from bending is an extremely interesting result - as it would suggest the internal electric field created by the ionic effect is significant. Could the authors please show this process, i.e. electrical generation by bending over 1 to 10 cycles (or however many cycles are required for this process) - this is will enable a significant benefit to future researchers. Was the short-circuit current in Figure 2 already showing this - if so, why didn't it increase with continued polarization?

Response:

We appreciate the reviewers’ recognition of this bend-self-polarization mechanism as well as the interest in it. Bend-self-polarization describes the process that the upper and lower PVDF layers achieve dipole alignment under the electric field of Nafion sandwich, which comes from the piezoionic effect under the bending stimulus. **Fig. 1d** in the original manuscript visually shows this process, as follows:

“Fig. 1 | Concept, design and properties of the piezo-ionic-electric electronics. d After repeated bending stimulation, the PNP electronics achieves piezoelectric activation, a particular process called bend-self-polarization.”

We strongly agree with the reviewer’s comment that there is a necessity to show this bending activation process in more detail, for completing the mechanism and benefiting the future researchers. Therefore, following the great advice from the reviewer, we have explored the bending cycles for self-poling through a large number of PNP samples. The corresponding test procedure and results are shown in **Supplementary Fig. 6** and have been added to the revised supplementary information.

Supplementary Fig. 6a is a diagram of the experimental setup for applying the bending stimulus to PNP devices. We flatten the PNP film device on two acrylic holders. The back and forth motion of the linear motor can apply the bending stimulation to the film at different degrees (curvatures). In order to accurately quantify the degree of bending behavior in terms of curvature, we utilized the film bending model for mechanical analysis, as shown in **Supplementary Fig. 6b**. The PNP film is tightly centered on the PU encapsulation layer (only one side is shown). When the ends are compressed by a distance ΔL , the bending of the PU will cause the PNP film to bend with the same curvature. Their out-of-plane displacement (h) can

be calculated as $h = A(1 + \cos(2\pi x_1/L))/2$, where A is the bending amplitude and L is the initial length of the film device (The length excluding the fixed portion at each end). Considering that the PNP film is at the center of the PU and its length is significantly smaller than the PU, the curvature (ω) of the PNP film can be calculated at $x_1 = 0$ (at the center of the device). It is given as $\omega = (-4\pi\sqrt{\Delta L/L})/L$ (Proceedings of the National Academy of Sciences of the United States of America 111, 1927-1932 (2014)). Accordingly, the curvature ω at a certain degree of bending can be calculated from the compression distance ΔL .

In order to quantify the bending cycle for bending self-polarization, we fix the bending frequency ($f = 1 \text{ Hz}$) and count the number of bending at three typical compression distances ($\Delta L_1 = 10 \text{ mm}, \Delta L_2 = 20 \text{ mm}, \Delta L_3 = 30 \text{ mm}$). **Supplementary Fig. 6c** show the actual photographs of the three states. Their curvatures (taking absolute values without considering directionality) increase with the compression distance as $\omega_1 = 0.192 \text{ mm}^{-1}$, $\omega_2 = 0.271 \text{ mm}^{-1}$ and $\omega_3 = 0.332 \text{ mm}^{-1}$, respectively. A group of 20 device-samples under each bending curvature was tested for number of bending. The statistical results are depicted in box-whisker plots and distribution curves, as shown in **Supplementary Fig. 6d**. The average number of bending for the three curvatures was 278, 125 and 31.75 times, respectively. It is clear that there is a negative correlation between the number of bending and the bending curvature. In fact, greater curvature implies a larger gradient of in-plane strain, which in turn induces a larger piezoionic electric field (Chemical Engineering Journal 482, 148988 (2024); Nano Letters 21, 5369-5376 (2021); Small 12, 5074-5080 (2016)). As a result, the polarization effect is subsequently more pronounced and the number of bends required to achieve the polarized state is decreased. Our experimental data is consistent with this theory. On the other hand, the standard deviation of the number of bending is greater at small curvatures. This is due to the fact that the experimental error becomes larger as the number of bends increases. By now, the number of bending in the three typical bending states is basically clear, adding important regularities to the development of the piezo-ionic-electric system.

Supplementary Fig. 6 Experiments about bending number for achieving bend-self-polarization. **a** Diagram of the experimental setup for applying the bending stimulus to PNP devices. **b** Bending mechanical model of PNP film-device. **c** Actual photographs of the three typical bending states the corresponding curvatures. **d** Statistical plot of the number of bending required to achieve polarization activation under three curvature conditions, where the statistical distribution of the number of bending is depicted by curves. Boxes and whiskers are drawn as the 25-75th and 5-95th percentiles, respectively. The inset shows the mean values of the number of bending and their relationship curves under three bending curvatures, and the overlapping part indicates the standard deviation.

A brief discussion to **Supplementary Fig.6** was also added to the revised manuscript:

“Moreover, a large number of samples have shown statistically that the number of bending required to achieve polarization exhibits a negative correlation with the bending curvature (Supplementary Fig. 6, details in Supplementary Note 1). At approximate limiting curvature (0.332 mm⁻¹), the average number of bending is about 32 times.” (page 8)

In addition, a more detailed discussion of the bending number experiment is added to the **Supplementary Note 1** in the revised supplementary information.

In addition, the short-circuit currents in **Fig. 2** mentioned by the reviewer are the results of the designed mechanism validation experiments and not a direct manifestation of the bend-self-polarization process. Their detailed process and significance is described below:

In order to validate the mechanism and further clarify the different roles of the two effects in the two stages, we designed the corresponding validation experiments in the original manuscript. By exploring the correspondence between the initial bending direction and the piezoelectric phase after activation, the dominant role of the piezoionic effect in Stage I is confirmed. In **Figs. 2b** and **d**, two unactivated PNP devices, each stimulated by bending moments in different directions (+Z and -Z), produce opposite current phases when stressed. Considering that the two devices are wired in exactly the same way in the piezoelectric test system, this difference can only come from a difference in the activation conditions (i.e., bending direction). Obviously, bending moments of different directions induce opposite directions of ion motion, which in turn produce opposite ion-based electric fields in the out-of-plane direction. Therefore, the overall polarization direction of the PVDF in the two devices is consequently opposite, which in turn generates short-circuit currents with opposite phases. This supports the decisive role of the piezoionic effect in Stage I.

On the other hand, for a PNP device that has been activated by bending in a certain direction and then bending it approximately in the reverse direction, there is no change in the phase of the current when it is pressed, as shown in **Figs. 2c** and **e**. This demonstrates the fact that dipoles of PVDF layers, which has been oriented by bending, is difficult to be reversed again by the opposite built-in ionic electric field, since such a reversal often requires a huge external directed energy. In other words, the direction of polarization of the bent self-polarized PVDF is relatively stable and is no longer easily affected by the ion-based electric field. Therefore, in Stage II, the piezoelectric effect responds to the pressure stably, constraining the piezoelectric ion effect and realizing the overall force-electric response.

In order to directly illustrate the significance of the above two sets of comparison experiments, we have modified the legend of **Figs 2b-e**, as follow:

“Fig. 2 | The unique piezo-ionic-electric working mechanism of the PNP electronics and its verification. **b** The process and principles of the validation experiments in Stage I. When the two PNPs were bent in opposite directions (-Z and +Z), the difference of internal microscopic evolution are compared schematically. **c** The process and principles of the validation experiments in Stage II. For the same PNP, bending in one direction (-Z) was followed by bending in the other direction (+Z), whose micro-evolution is also shown. **d** Plot of opposite short-circuit current phases, supporting the decisive role of the piezoionic effect in the bend-self-polarization. **e** Plot of unchanged short-circuit current phase, reflecting the leading role of piezoelectric effect in the overall response process.”

Comment 4: Given the outstanding result of self-poling through bending, it raises the question of relative contribution between the piezoelectric polymer compared to the confined nafion itself? Is it possible to provide control experiments where nafion is sandwiched between two non-piezoelectric polymers (i.e. sheets of PTFE for a fluoropolymer comparison or another appropriate polymer) - again, this deconvolution of the magnitude of the effects would highlight the role of synergistic performance of piezo-ionic effects as opposed to simply ionic effects.

Response:

Thank the reviewer for this constructive and meaningful comment. We strongly agree that such controlled experiment will play an important role in deconvolution of synergistic effects. Following the reviewer's suggestion, we successfully prepared PTFE sandwiched films with Nafion interlayer, which was shown not to possess force-electric coupling properties. The corresponding discussion and results are as follows:

Bend-self-polarisation is a process in which an ionic polymer sandwich (Nafion in this work) is stimulated by bending to generate a built-in ion-based electric field, which in turn induces dipole orientation alignment in the upper and lower piezoelectric polymer layers (PVDF in this work). Thus, piezoionic effect is the prerequisite and the piezoelectric effect is the consequence in this process, and both are necessary.

The corresponding control experiments have been added as **Supplementary Fig. 9** in revised supplementary information. The good homogeneity of the PTFE-Nafion-PTFE film is demonstrated in the Photograph of **Supplementary Fig. 9a**. PTFE also bonds stably to Nafion, attributed to the fluorocarbon backbone of Nafion. The FTIR spectra measured in total reflection (ATR) mode show the film has three characteristic absorption peaks of PTFE at 1200 cm^{-1} , 1145 cm^{-1} and 639 cm^{-1} wavenumbers (Polymers 11, 1629 (2019)) (Supplementary Fig. 9b). After that, the film was sputtered with Ag electrodes and was encapsulated with PU, as shown in **Supplementary Fig. 9c**. However, when the device was applied the same bending

process as the PNP, the piezoelectric test data indicated that it does not possess force-electric coupling characteristics (**Supplementary Fig. 9d**). This result fully corroborates the crucial role of piezoelectric effect in the Stage II, i.e., the dynamic piezoionic-coupled piezoelectric response.

Meanwhile, we have added relevant notes at appropriate places in the revised manuscript:

“It is worth mentioning that if PVDF is replaced with polytetrafluoroethylene (PTFE), a non-piezoelectric polymer, the piezoelectric response after bending self-polarisation cannot be achieved (**Supplementary Fig. 9**).” (page 11)

Supplementary Fig. 9 Control experiments on PTFE sandwich films with Nafion interlayers. **a** Photograph of the PTFE-Nafion-PTFE film. The scale bar is 10 mm. **b** FTIR in attenuated total reflectance (ATR) mode for this film. The results show the PTFE characteristic peaks¹³. **c**

Structural diagram of the PTFE-based device, for piezoelectric performance testing. **d** The corresponding piezoelectric test data showed no piezoelectric properties of this PTFE-based device.

In addition, as the reviewer states, the relative contributions of the piezoelectric polymer (piezoelectric effect) and the confined Nafion (piezoionic effect) in the overall piezo-ionic-electric working mechanism require clarification. For this purpose, we have previously designed comparative validation experiments in the original manuscript:

By exploring the correspondence and changes between the bending direction and the phase of the piezoelectric signal after activation, the dominant roles played by the piezoionic effect in Stage I, and the piezoelectric effect in Stage II, respectively, are confirmed. The detailed process and principles have been mentioned in the *Response to the Comment3*.

More notably, we designed short-circuit current phase calibration experiments in the revised manuscript to directly demonstrate the respective roles of the two effects in the overall mechanism. This relevant process has been added as **Supplementary Fig. 10** in revised supplementary information, as follow:

Supplementary Fig. 10 Current phase calibration to assist in verification of bend-self-polarization dominated by piezoionic effect. **a** Electrometer synergized with signal generator for calibrating the short-circuit current phase, and the corresponding equivalent circuit. **b** The signal generator produces a square wave signal as a power source. **c** The square-wave short-circuit current detected by the electrometer is negative value. **d** After bending the PNP device in the +Z direction, according to the principle of bend-self-polarization, the tensile surface is equivalent to the positive pole of the power supply, and the compressed surface is the negative pole (when pressed). **e** With the wire connections consistent with calibration circuit, the device produces a negative current when stressed, which is in accordance with the results of the calibration circuit.

Discussion of **Supplementary Fig.10** was also added to the revised manuscript:

“In addition, we supplemented the short-circuit phase calibration experiments (Supplementary Fig.10 and Supplementary Note 3 for details). According to the experimental results, for the PNP device, the tensile side during bend activation can be equated to the positive pole of the power (when pressurized). This is consistent with Nation’s theory of piezoionic effect²⁶. It further validates the piezo-ionic-electric mechanism and the respective roles of these two effects in it.” (page 11)

A more detailed discussion is likewise added to the **Supplementary Note 3** in revised supplementary information.

Comment 5: Can the authors please describe the piezoelectric testing methodology used in detail. The set-up described in Figure S6 appears to show that contact-separation mode is used, which induces significant triboelectrification in addition to piezoelectric and ionic effects? These methods need to report the force applied for testing, along with the waveform used—unless these methods are adequately described it is impossible to rule out alternative contributions to the charge generation mechanism (see: <https://onlinelibrary.wiley.com/doi/full/10.1002/aelm.202400019>; <https://www.nature.com/articles/s41467-022-29087-w>; <https://onlinelibrary.wiley.com/doi/abs/10.1002/adma.202002979>;)

Response:

We thank the reviewer for this professional advice. As commented, all of our piezoelectric signal tests fall into the contact-separation mode. We followed the suggestion and supplemented the force loading curves (waveforms) used for testing. It helps to exclude interference from triboelectric signals to ensure the accuracy of piezo signal data.

This relevant results have been added as **Supplementary Fig.21** in revised supplementary information. Firstly, we visually depict the parts and materials composition of the contact-separation test platform in **Supplementary Fig. 21a**. According to the analyze of the reference (Nature Communications 13, 1391 (2022)), the acrylic round for compression and the encapsulating material polyurethane may generate triboelectric signals due to the contact-separation process. To rule out this possibility, we compared the force loading curve (**Supplementary Fig. 21d or e**) with the charge curve (**Supplementary Fig.21b**) and the voltage curve (**Supplementary Fig. 21c**) in the same time domain. The results show that there is no phase difference between the force curve and the electrical curve. It is when the acrylic starts to touch the device that the charge curve (or voltage curve) starts to take a plunge (or increase). And when the force disappears, so does the electrical signal. There is no significant signal change during the non-contact period of gradual approach and separation. This force-electric synchronization proves that there is almost no triboelectric signal during the contact separation piezoelectric test. In addition, in the forward and reverse connection experiment (**Supplementary Fig. 20**) already in the manuscript, the current peaks were almost identical after the phase change. Again, it was demonstrated that the measured electrical signal was piezoionic-coupled piezoelectric signal.

Supplementary Fig. 21 Exclusion of triboelectric signals using force loading curves. **a** Schematic diagram of the contact separation piezoelectric test platform used in this work. **b** Transferred charge curve and **d** corresponding force loading curve. **c** Open circuit voltage curve and **e** corresponding force loading curve.

Discussion of **Supplementary Fig. 21** was also added to the revised manuscript:

“First, electrodes of the PNP electronics were connected to the electrometer first in the forward direction and then in the reverse direction, and it was found that the current also shifted in phase and have almost the same peak current (Supplementary Fig. 20). Further, there is no phase difference between the force loading curve and the electrical signal curve in the same time domain, even in contact-separation testing mode (Supplementary Note 6 and Supplementary Fig. 21).” (page 19)

Considering that we obtained the important parameter of the piezoelectric coefficient d_{33} using the direct piezoelectric charge method in the contact-separation mode, we further obtained the accurate transferred piezoelectric charge using the compressed balance analysis (CBA) method proposed in the above reference.

The analysis principle is shown in **Supplementary Fig. 16**. We define the real piezoelectric transfer charge as Q_p . The transferred charge in the Acrylic plate is Q_2 , and $-Q_1$ is the transferred charge in the polyurethane encapsulation layer. And q (or q') is the total charge transfer including piezoelectricity and triboelectricity, which is also the charge measured by the electrometer. In **Supplementary Fig. 16a**, during the compression phase, all charges obey the law of electrostatic balance. Therefore, they have the following relationship:

$$Q_2 - Q_1 + q - Q_p = 0$$

After that, flip the PNP device so that its positive polarization side is facing out (**Supplementary Fig. 16b**). The electrostatic balance becomes:

$$Q_2 - Q_1 - q' + Q_p = 0$$

Combining the above two equations, we have:

$$Q_p = \frac{q + q'}{2}$$

By testing q and q' at different forces, the piezoelectric transfer charge Q_p can be obtained. Therefore, the direct piezoelectric charge method enables the calculation of an accurate d_{33} value that excludes triboelectric interference. Under a gradient force of 1-35 N, the total charge transfer curves of the PNP device with the negative and positive polarization surfaces facing outwards is shown in **Supplementary Figs.16c** and **d**. Sequentially, q and q' were analyzed as in **Supplementary Figs. 16e** and **f** to obtain Q_p at each force. And a linear fit was performed to obtain the value of d_{33} under the compressed balance analysis method. The results (**Supplementary Fig.16g**) show that the d_{33} obtained by this method is 80.39 pC N^{-1} , which is nearly consistent to the previous method of direct average analysis (80.70 pC N^{-1} , averaging three measurements with only one polarisation surface facing outwards). Also the piezoelectric

transfer charges at each force obtained by the two methods are close to each other (**Supplementary Fig. 16h**). This again demonstrates that the effect of triboelectric signals in the contact- separation piezoelectric test method in this work is almost negligible. In other words, the piezoelectric signals in this work have sufficient accuracy.

Discussion of **Supplementary Fig.11** was also added to the revised manuscript with appropriate reference citations:

“In addition, it is necessary to exclude the interference of triboelectric charges during the testing process⁵⁰. We used the recently proposed compressed balance analysis method⁵¹ as a correction for piezoelectric charge testing. The results show that the d_{33} value of the two methods are quite similar (For details, see Supplementary Note 6 and Supplementary Fig. 16).” (page 15)

In addition, above detailed discussions have been centrally added as **Supplementary Note 6** in the revised supplementary information for better presentation.

Supplementary Fig. 16 Testing real piezoelectric charges by compressed balance analysis method for d_{33} evaluation. Principles of compressed balance analysis, including **a** negative polarization direction and **b** positive polarization direction after flipping. The total charge

transfer curves of the PNP device with the **c** negative and **d** positive polarization surfaces facing outwards. The total transferred charge measured from the **e** negative and **f** positive polarization surfaces, respectively, under a force of 25 N. **g** Force-charge curves obtained based on the compressed balance analysis method and comparison with existing direct average method. **h** A case-by-case comparison of the piezoelectric transferred charges measured under each force by the two methods.

Comment 6: The authors state force applied in Figure 4 ranged from 1 to 35N - were these from 0 to 1 up to 0 to 35N (i.e. contact-separation mode) or was a pre-load force used?

Response:

Thanks again the reviewer for this expert comment about piezoelectric test methods. All of the piezoelectric test modes in this work are contact-separation mode. In the tests of open-circuit voltage (**Fig. 4a**), short-circuit current (**Fig. 4b**), and transfer of charge (**Supplementary Figs. 16c and d**), the applied gradient force is 0 to 1 up to 0 to 35 N.

The pre-load force method proposed by the reviewer can effectively avoid the interference of friction signals during the contact-separation process. But as we response in *Comment 5*, there is no need to worry about signal interference in the piezoelectric test method of this work. The piezoelectric signals measured by contact-separation are fully credible, both in terms of the correspondence of the signal phase (**Supplementary Fig. 20**) and the accuracy of the real piezoelectric transfer charges (**Supplementary Fig. 16**).

The relevant discussion in the manuscript has been modified in order to more accurately describe the pattern of applied gradient forces:

“The voltage and current output of the PNP under gradient pressure (1-35 N, contact-separation mode) were subsequently tested.” (page 19)

Comment 7: Can the authors please normalize their power and energy values to either per unit area or per device volume to enable comparison to prior works?

Response:

Thank the reviewer for this constructive comment. We fully agree that normalizing power values to power per unit area will facilitate future peer comparisons. In this work, we select the effective area of the electrode for power normalization. The Ag electrodes of all devices, are circles with radius $R = 6.0 \text{ mm}$ and effective area $S = 113.1 \text{ mm}^2$. Therefore, the maximum peak power of both PNP as well as PPP has been recalculated in the revised manuscript to be 34.66 mW m^{-2} and 0.057 mW m^{-2} , respectively. It is worth emphasizing that the data were measured at a pressure of 20N (176.9 kPa). The effect of the magnitude of the external force must not be ignored in the comparison.

Corresponding figures and text descriptions have been modified. The revised power versus external resistance plots for PNP and PPP are as follows:

“Fig. 4 | Systematic testing and analysis on PNP piezo-ionic-electric sensing and energy harvesting characteristics. **g** Instantaneous peak power per unit area with external resistance.....”

“Supplementary Fig. 23 Output test of the PPP with external resistance. **b** Instantaneous peak power per unit area with external resistance.”

Comment 8: Did the two PVDF layers contact at high applied pressure? If so, the displacement of the liquid may explain the decreased performance above 220 kPa.

Response:

We thank the reviewer’s meticulous consideration. According to the cross-section SEM image of the PNP film (**Fig. R3**), the hot-pressed Nafion layer still maintains a thickness of 9 μm, which is not much different from the upper and lower PVDF layers, which are each about 20 μm thick. Meanwhile, considering that Nafion has the fluorocarbon backbone of polytetrafluoroethylene (PTFE), it has a certain level of tensile strength (23 MPa, *ASTM D 882* standard test method, data from Chemours Company). Therefore, we do not think that the Nafion layer would be overly compressed at higher pressures above 220 kPa, resulting in the upper and lower PVDF layers coming into contact with each other, resulting in proposed unnatural displacement of the liquid or possible damage to the ion-electron interfaces.

In this issue, we consider that it is the limited proton concentration (or limited proton mobility) in the Nafion layer that constrains the ion-electron polarization response of the PNP under higher voltages. As a result, the device gradually became saturated in interfacial polarisation under the high-voltage condition and cannot maintain the same linear response as under low-pressure condition.

Fig. R4 Thickness of layers in PNP film obtained from cross-section SEM image. Coloring the PVDF layer orange, the Nafion interlayer blue.

We further explain the main reason for the sensitivity segmentation in the revised manuscript:

“The limited protons content in the Nafion interlayer constrains the saturation tendency of the interfacial polarization, which may be the main reason for the segmented sensitivity of the PNP.”

Comment 9: In the introduction, can the authors describe the advantages of Piezoelectric-Ionic energy harvesters compared to piezoelectric-triboelectric energy harvesters (see: <https://doi.org/10.1021/acsami.1c04489>; <https://pubs.acs.org/doi/full/10.1021/acsaem.3c01196>) - these appear to be the direct comparative technology which could be produced via an air gap in the PVDF film.

Response:

We would like to thank the reviewer for this valuable suggestions. As a typical hybridized self-powered sensor in recent years, piezoelectric-triboelectric combines both the flexibility of piezoelectric devices and the high output of triboelectric devices. These two works given by the reviewer have constructed high-performance piezoelectric-triboelectric hybridized energy harvesters by simultaneously using piezoelectric materials, PVDF with high β -phase content (ACS Applied Energy Materials 6, 9300-9306 (2023)) and porous PVDF hollow fibers (ACS Applied Materials & Interfaces 13, 26981-26988 (2021)), as triboelectric materials. However, as another new type of hybridized self-powered sensor, our piezo-ionic-electric electronics make up for the two shortcomings of the above piezoelectric-triboelectric devices.

On the one hand, piezoelectric materials in piezoelectric-triboelectric devices often require additional high-voltage polarization processes to enhance the piezoelectricity. Examples are corona polarization or electrostatic spinning in the two aforementioned works. This undoubtedly complicates and risks the preparation method, which is not conducive to large-scale manufacturing and energy-efficient manufacturing of piezoelectric parts. In contrast, our device is capable of piezoelectric self-polarization through a simple bending behavior with the help of a built-in piezoionic electric field.

On the other hand, piezoelectric-triboelectric hybrid devices cannot avoid the complexity of separate structures that are not a tightly integrated whole. Instead, our piezo-ionic-electric devices have a straightforward structure that enables sensing with a tightly bonded single film less than 100 μm thick. This unity undoubtedly gives our device the ability to work stably for long periods of time, while expanding the usage scenarios (e.g., wearable electronics).

Therefore, as suggested by the reviewer, we have added a relevant discussion in the introduction section and cited the two above references:

“Combining the two effects enables the development of self-powered hybrid systems with simple structure and novel mechanism, will be greatly improved compared to the existing piezoelectric-triboelectric hybrid generators³²⁻³⁴.” (page 4)

Comment 10: While the work focusses on PVDF, the authors should highlight the potential of piezoelectric-ionic effects enable non-fluoropolymer-based energy harvesters (see: <https://doi.org/10.1002/sml.202311570>)

Response:

Thank the reviewer for this constructive suggestions. We strongly agree with the attractiveness of non-fluoropolymers in terms of environmental friendliness. Expanding the piezo-ionic-electric system to non-fluorinated polymers bases, not just per- and polyfluorinated alkyl substances (PFAS) such as PVDF bases, would be very significant in terms of green and sustainable energy harvesting. Fortunately, most non-fluorinated piezoelectric polymers, such as glycine, a bio-derived polymer, have similar principles and modes of operation to PVDF, which certainly increases the potential for this sustainable development (Small 20, 2311570 (2024)).

Therefore, we have additionally emphasized this potential in the discussion section:

“We have confidence that the paradigm of piezo-ionic-electric electronics as well as its intrinsic interfacial ionic-electronic effect will have a significant impact on the innovative development and valorized utilization of polymer-based, including PVDF-based and more non-fluoropolymer-based, piezo-generators.” (page 25)

Also recognizing the importance and specificity of non-fluoropolymer piezoelectric materials in the overall flexible piezoelectric electronics, we have cited the above reference in the beginning of the introduction:

“Prompted by the continuous pursuit of new renewable energy and the new generation of wearable devices, the flexible piezoelectric electronics have gained significant momentum in recent years¹⁻³.” (page3)

List of Responses to the Reviewers' Comments

We would like to thank the reviewers for their recognition of this manuscript and valuable comments and suggestions, which further greatly improves the quality of our manuscript. Following these comments and suggestions, we have made careful revisions to our previous manuscript and supplementary information (marked in red color), and provide response to the comments point-by-point as follows:

Reviewer #2 (Remarks to the Author):

The authors have fully addressed all comments raised by us. Therefore, the revised manuscript can be published in the Nature Communications journal without any correction.

Response:

We are very grateful to the reviewer for the high recognition of our work, which will motivate us to explore further in this field.

Reviewer #3 (Remarks to the Author):

The authors have performed significant new experiments and reworked their manuscript which has resulted in a dramatic improvement in the overall quality of the manuscript. In particular, the performance of experiments with PTFE-nafion-PTFE showing a negative result provides significant evidence in support of the claims within the manuscript - I thank the authors for taking the time to perform these experiments and produce such clear and significant changes. The manuscript now contains significant novelty, clear methodology, and strong control experiments - and I believe the manuscript is suitable to be published in Nature Communications.

Response:

We would like to thank the reviewer for their high recognition of this work. We are very grateful for the valuable and professional comments previously made, which were significant and

greatly improved the quality of the manuscript. Three new minor comments are also important and interesting. The response to these comments point by point is as follows:

Comment 1: please add relevant references to Table S3: MXene (<https://doi.org/10.1038/s41467-021-23341-3>); CNT (<https://doi.org/10.1039/C9EE03059J>).

Response:

We thank the reviewer for literature recommendations, which have made the references in this manuscript more relevant and more convincing. We have added the d_{33} data of PVDF-TrFE/MXene literature (<https://doi.org/10.1038/s41467-021-23341-3>) to Supplementary Table 3 and also added the corresponding bar in Supplementary Fig. 17 (see below). The second reference about the PVDF-TrFE/CNT (<https://doi.org/10.1039/C9EE03059J>) is also important for this paper. Unfortunately, we are very sorry that this literature has chosen pm/V as the unit for d_{33} , so it cannot be compared with the literatures in the Supplementary Table 3 and Supplementary Fig. 17, whose d_{33} unit is pC/N. However, considering the importance of this literature in the piezoelectric composite materials field and its high relevance to our work, we have cited it in the opening part of the Introduction. As well, we have cited the first reference above in this location. The corresponding changes are as follows:

“To break through the above bottleneck, PVDF is classically added with polar inorganic fillers like ceramic nanoparticles^{12,13}, carbon nanotubes¹⁴ and MXene nanosheets^{15,16}.” (page 3)

“14. Shepelin, N. A. et al. Printed recyclable and self-poled polymer piezoelectric generators through single-walled carbon nanotube templating. *Energy Environ. Sci.* **13**, 868-883 (2020).” (page 29)

“15. Shepelin, N. A. et al. Interfacial piezoelectric polarization locking in printable Ti3C2Tx MXene-fluoropolymer composites. *Nat. Commun.* **12**, 3171 (2021).” (page 29)

Supplementary Table 3

Comparison of d_{33} values between PNP films and recent PVDF-based piezoelectric systems.

Piezoelectric functional material	Self-polarizing or not	d_{33} (pC N ⁻¹)	References
PVDF/Nafion	Yes	80.7	Our work
PVDF/KNN	No	12	14
PVDF-TrFE	No	21	15
PVDF-TrFE/PC	No	49.1	16
PVDF	No	62	10
PVDF/CNTs	Yes	9.4	17
PVDF/Gly-MoS ₂	Yes	24.9	18
PVDF/BTO	Yes	51.2	11
PVDF-TrFE/MXene	Yes	52	19
PVDF/MXene	Yes	63.3	20

Supplementary Fig. 17 Comparison of d_{33} values between PNP films and recent PVDF-based piezoelectric systems^{10,11,14-20}.

Comment 2: the space charge of the structure demonstrated by COMSOL multiphysics Figure S19 should have a colour scale bar to denote the magnitude of the fields.

Response:

Thanks for the reviewer’s careful check. We apologize for our omissions in the COMSOL Multiphysics figures. We have supplemented the color scale bar to indicate the magnitude of the fields, as shown below. This simulation figure semi-quantitatively compares the potentials of the three models of PPP, Nafion, and PNP.

Supplementary Fig. 19 The simulations of space charge distribution of models of a PPP, b Nafion and c PNP by COMSOL Multiphysics.

Thanks again to the reviewer for the patience in checking the manuscript.

Comment 3: the layer-by-layer space charge demonstrated in Figure S19 suggests the working mechanism of the device is similar to layered 3D TENGs (<https://onlinelibrary.wiley.com/doi/full/10.1002/aesr.202300259>), this doesn't need to be addressed here - but it's an interesting comparison.

Response:

We thank the reviewer for this interesting view, which has inspired our subsequent research, especially about the piezo-ionic-electric working mechanisms. We strongly agree that there is common ground between the layered 3D TENG and our piezo-ionic-electric generator. Both rely on the generation of charges at the component interfaces to produce electrical output. The difference is in the way of charge generation. The former originates from contact-separation processes between different components (e.g., sliding between small and large fibers in the given literature), while the latter originates from interactions between ion-based and electron-based electric fields under pressurized conditions. This interesting comparison is instructive for our further work.